# Accounting for environmental and fishery management factors when standardizing CPUE data from a scientific survey: A case study for *Nephrops norvegicus* in the Pomo Pits area (Central Adriatic Sea)

Matteo Chiarini [1,2]*, Stefano Guicciardi[1], Silvia Angelini [1,3], Ian D. Tuck[4],
Federica Grilli[1], Pierluigi Penna [1], Filippo Domenichetti[1], Giovanni Canduci[1],
Andrea Belardinelli[1], Alberto Santojanni[1], Enrico Arneri[1], Nicoletta Milone[5],
Damir Medvešek[6], Igor Isajlović [6], Nedo Vrgoč[6], Michela Martinelli [1]

1 National Research Council–Institute of Marine Biological Resources and Biotechnologies (CNR IRBIM),
Ancona, Italy, 2 Department of Biological, Geological and Environmental Sciences, University of Bologna
(UNIBO), Bologna, Italy, 3 Fano Marine Center, The Inter-Institute Center for Research on Marine
Biodiversity, Resources and Biotechnologies, Fano, Italy, 4 National Institute of Water and Atmosphere
(NIWA), Auckland, New Zealand, 5 Food and Agriculture Organization (FAO), Rome, Italy, 6 Institute of
Oceanography and Fisheries (IOF), Split, Croatia

* matteo.chiarini5@unibo.it

journal.pone.0270703

Ricerca - Stazione Sperimentale per lo Studio delle
Risorse del Mare, ITALY

## Abstract

Abundance and distribution of commercial marine resources are influenced by environmental variables, which together with fishery patterns may also influence their catchability. However, Catch Per Unit Effort (CPUE) can be standardized in order to remove most of the variability not directly attributable to fish abundance. In the present study, Generalized Additive Models (GAMs) were used to investigate the effect of some environmental and fishery covariates on the spatial distribution and abundance of the Norway lobster *Nephrops norvegicus* within the Pomo/Jabuka Pits (Central Adriatic Sea) and to include those that resulted significant in a standardization process. *N. norvegicus* is a commercially important demersal crustacean, altering its catchability over the 24-h cycle and seasons according to its burrowing behavior. A historically exploited fishing ground for this species, since 2015 subject to specific fisheries management measures, is represented by the meso-Adriatic depressions, which are also characterized by particular oceanographic conditions. Both the species behaviour and the features of this study area influence the dynamics of the population offering a challenging case study for a standardization modelling approach. Environmental and catch data were obtained during scientific trawl surveys properly designed to catch *N. norvegicus*, thus improving the quality of the model input data. Standardization of CPUE from 2 surveys from 2012 to 2019 was conducted building two GAMs for both biomass and density indices. Bathymetry, fishing pressure, dissolved oxygen and salinity proved to be significant drivers influencing catch distribution. After cross validations, the tuned models were then used to predict new indices for the study area and the two survey series by means of informed spatial grids, composed by constant surface cells, to each of which are associated

**Data Availability Statement:** Datasets including the oceanographic parameters used in this study to develop the GAM approach have been published in the SEANOE data repository and are available at the following links: https://doi.org/10.17882/85925 (Penna et al. 2022a) and https://doi.org/10.17882/86456 (Penna et al. 2022b). Furthermore maps related to the specific bottom values used are presented in the Supporting information (S6–S11 Figs in S1 File). The catch data and fishing activity information related to each trawl haul carried out during the considered surveys and used to calculate CPUE values were for the most part collected under stated confidentiality agreements between the Italian Ministry of Agricultural, Food and Forestry Policies (MIPAAF) and the Institute for Marine Biological Resources and Biotechnologies of the National Research Council (CNR IRBIM). A small portion of the dataset was also collected under similar conditions with the Croatian Institutions. The authors received formal authorization to use the data for scientific purposes but no special access privileges were granted. All data are stored at CNR IRBIM Ancona and can be shared on reasonable request upon prior authorisation by MIPAAF; the interest of qualified researchers may be expressed to pemac1@politicheagricole.it. Furthermore maps for normalised CPUE values obtained for each considered trawl haul are presented in the Supporting information (S2–S5 Figs in S1 File)

**Funding:** Funding was provided for the collection and processing of the data used in this study by the Direzione Generale della Pesca Marittima e dell'Acquacoltura of the Italian Ministry of Agricultural, Food and Forestry Policies (through a series of agreements with the Institute for Marine Biological Resources and Biotechnologies of the Italian National Research Council; CUP J52I15003990001, J53C17000540001 and J41F19000080001), the RITMARE Flagship Project of the Italian Ministry of University and Research and the FAO AdriaMed regional project. Michela Martinelli received partial financial support from the CNR Short-Term Mobility Program 2019.The authors received no specific funding for this work.

**Competing interests:** The authors have declared that no competing interests exist.

average values of environmental parameters and specific levels of fishing pressure, depending on the management measures in place. The predictions can be used to better describe the structure and the spatio-temporal distribution of the population providing valuable information to evaluate the status of such an important marine resource.

## Introduction

Information about the distribution and abundance of marine species is usually derived from fishery-dependent data (e.g. sampling on board commercial vessels) or fishery-independent data (e.g. scientific surveys at sea) [1]. Catch Per Unit of Effort (CPUE), or catch rate, can be used as an index of abundance and it is the primary source of information for many of the most valuable and vulnerable commercial fishery resources [2] including several crustacean species [3, 4]. However raw CPUE, intended as the total catch divided by an observable measure of associated effort, is rarely proportional to the real abundance of a resource over time and space [5], because numerous factors can affect catch rates [6]. Indeed, the CPUE index depends on the abundance of a resource, the fishing effort and also the catchability (intended as the fraction of an available resource that is captured by one unit of effort) [6, 7]. Catchability can be assumed as constant or changes in catchability can be modeled and removed from the index [8]. Indeed, catchability is often spatiotemporally affected by environmental, biological, and management (such as fishery technological and/or management strategy changes) factors [2, 9, 10]. The process of removing the effect of factors that bias catch rate as an index of abundance is commonly referred to as CPUE standardization [11] and it allows obtaining more accurate estimates of abundance indices [12, 13] and related standard errors [14, 15]. In fact, fishery-dependent raw CPUE commonly vary over both space and time showing coverages mainly driven by fisheries patterns [16]. Even time series obtained from scientific surveys may suffer for gaps in space and time (e.g. missing years, sampling stations allocation, etc..) as well as for discontinuities in the sampling intensity or strategy (e.g. change of protocol, vessel, gear used). Therefore, it would be relevant to remove factors other than abundance that may influence CPUE variability before using an index as an indicator of population size [7, 17].

Moreover, ecological processes (both exogenous and endogenous) can affect the species distribution and density [18–20]. Such a spatio-temporal dependence has been traditionally incorporated into CPUE standardization using spatial grids to predict the spatial distributions and relative abundances of marine species [e.g. 21, 22]. Therefore, to accurately estimate the total population, a standardization process should first of all account for spatial and temporal differences in sampling rates, for example when the spatial pattern of the observation is not adequate [23] or when statistical dependence arises from biological or sampling characteristics (e.g shoals movement and extraordinary catch events) [24]. Second, the standardization should account for the effect of covariates which could have an impact on the catch rates over time and space [25]. Environmental conditions are among the factors that control the spatio-temporal distribution of fish populations, therefore it is crucial to identify their relationships with catch rates [26]. In addition, changes in fishing effort due, for example, to the establishment of a temporary fishing ban or a no-take zone, may result in changes of catch rates in time [27] and space (e.g. buffer areas [28, 29]). However, the standardization of a catch rate time series must be designed to include only explanatory variables that significantly influence the dependent variable (i.e. CPUE [30]). Too few explanatory variables will cause variation in

catch rate to be wrongly attributed to the time series, while too many explanatory variables will over-fit the model [11].

Historically, many efforts have been made to solve the difficulties associated with CPUE standardization [7] promoting the flexibility and the availability of well-tested and user-friendly tools such as Generalized Linear Models (GLMs) and Generalized Additive Models (GAMs) to perform calculation [31]. However, while considering the nonlinearity of predictors there is evidence that statistical models such as GAMs perform better than GLMs [32] even if the survey area is not well covered due to the lack of sampling locations or biased designs [33]. GAMs proved to be also helpful to understand the environmental processes underlying species distributions [34]. Furthermore, with the aim to produce robust abundance indices with associated standard errors [34], more sophisticated approaches such as Vector Autoregressive Spatio-Temporal (VAST) model and Boosted Regression Tree (BRT) were also recently implemented [35, 36].

Within the Mediterranean basin, the Norway lobster, *Nephrops norvegicus* (Linnaeus, 1758) (hereafter referred to by genus alone), represents one of the main commercial species in terms of value and is mainly targeted by bottom trawlers [37]. This demersal species lives at depths from around 30 m to over 400 m [38, 39] with a preference for muddy grounds allowing the formation of its characteristic burrows [40]. This species is mainly captured by bottom trawlers when it emerges from its burrows [41–43]. Several studies provided evidence of a rhythmicity in the burrow emergence pattern showing peaks that vary in time depending on depth [44–46] and other ecological and demographic factors (i.e. food availability, size, sex, reproductive status [47–49]). Literature put in evidence how changes in environmental parameters, such as dissolved oxygen [40], salinity [50] and bottom temperature [51], proved to be fundamental drivers influencing life cycle and emergence behaviour of wild populations of *Nephrops*. Therefore, the emergence rhythm may cause marked fluctuations in CPUE over the 24-h cycle [39, 52, 53] and seasons depending on sex (females rarely leave their burrows during the egg-bearing period [40, 54]), which means that the trawl fishery exploits the population selectively and in a different manner during the year. As suggested by Sardà and Aguzzi, accounting for such an availability of *Nephrops* to trawlers, not only according to the time of the day but also at a seasonal scale, may lead to more reliable estimates of population abundance [55]. Hence these differences in *Nephrops*' CPUE over different time scales (i.e. time of the day and season) should be considered directly within the standardization process, when possible.

In the Adriatic Sea, *Nephrops* is subjected to high fishing pressure from fleets of different countries [56]. Relevant concentrations of this species occur off Ancona, in the Pomo/Jabuka Pits and in the Velebit Channel [53, 57–62]. There is evidence that *Nephrops* in the Adriatic Sea is also characterized by distinct subpopulations which should be considered as separate units [63, 64] especially from the fishery management point of view [65]. According to a recent Scientific Technical and Economic Committee for Fishery report [66], in the Adriatic Sea this species is considered overexploited; the fishery-independent data used to implement this stock assessment derived from scientific surveys not specifically targeting *Nephrops*. However this type of data source was already defined not fully appropriate to overcome the general issues related to the catchability of this species, and the need for more detailed information from surveys specifically targeting this resource, in order to support an accurate *Nephrops* management, was already stated [65].

A sustainable use of fishery resources should be fostered by an ecosystem approach to fisheries management [67, 68]. To move towards this, it is crucial to understand the impact of fisheries and environmental factors on benthic communities and their dynamics [69, 70]. Angelini et al. developed an ecosystem model for the Pomo/Jabuka Pits and highlighted the need of improving it including environmental variables thus considering the influence of the

hydrographic processes within the meso-adriatic depression on *Nephrops* distribution [71]. Furthermore, starting from 2015, the Jabuka/Pomo area has been the object of a series of management measures, which have greatly modified the spatial patterns of the fishing effort [28, 72].

Focusing on data obtained from two annual dedicated *Nephrops* surveys carried out in the Pomo/Jabuka Pits, the present study proposes the use of GAMs as tools to: i) investigate how the environmental variables and the different management strategies can influence abundance and distribution of *Nephrops*, ii) produce standardized CPUEs, to be used as reliable inputs for population dynamic and ecosystem models.

## Materials and methods

### The study area

The study area defined in Fig 1 is located in the Central Adriatic Sea and includes the three meso-Adriatic depressions collectively known as Pomo/Jabuka Pits, that altogether have a maximum depth of 270 m [73] and a seabed mainly composed by fine muddy substrates [74].

The depth range and sediment composition make the Pomo/Jabuka Pits an ideal habitat for *Nephrops* [38–40] whose subpopulation is here characterized by high densities of individuals smaller than those dwelling in nearby areas (e.g. off Ancona) [57, 63]. Furthermore, this area was also identified as a nursery ground for the European hake, *Merluccius merluccius* (Linnaeus, 1758), which is another important commercial demersal species in the Adriatic Sea [77, 78]. Owing to the presence of essential habitats in a region exploited by both Italian and Croatian fisheries, the Pomo/Jabuka Pits was subjected to several discussions regarding the possibility of establishing an area closed to fisheries to protect heavily exploited demersal stocks [79, 80]. The first partial closure of trawling activities was approved in July 2015 (D.M. 03/07/2015 and D.M. 20/07/2016; N.N. 20/07/2015 and N.N. 22/07/2016); then several decrees changed the fishing restrictions over time and space (D.M. 19/10/2016; D.D. 7/12/2016; D.M. 01/06/2017; N.N. 17/05/2017). Finally in 2018, a Fishery Restricted Area (FRA) was established identifying a fishery ban zone (S1 Fig in S1 File, zone A) and two buffer zones where the fishing is limited to a specific number of authorized vessels and fishing days (S1 Fig in S1 File, zones B and C) [81–83].

### Data collection

The analyzed data originate from two fishery-independent surveys carried out annually within the Pomo/Jabuka Pits: (i) the spring "Under Water TeleVision" (UWTV) survey, performed jointly by the National Research Council—Institute of Marine Biological Resources and Biotechnologies (CNR IRBIM) of Ancona (Italy) and the Institute of Oceanography and Fisheries (IOF) of Split (Croatia) in the whole study area; (ii) the autumn "ScamPo" experimental trawl survey, carried out by CNR IRBIM of Ancona western of the Adriatic midline [84]. The "UWTV" (hereafter referred to as spring) survey principally aims to quantify *Nephrops* burrows through video analysis of seabed footage [85, 86] but during this also experimental trawling activities are carried out [63, 72, 87]; these latter data on trawling were those considered in the context of this study. Instead, the "ScamPo" (hereafter referred to as autumn) survey is mainly meant to monitor the effects of the management measures on commercially relevant demersal species (including *Nephrops*) [72]. The time series considered within this study ranges from 2012 to 2019 for the spring surveys, and from 2015 to 2019 for the autumn ones.

As suggested by Kimura and Somerton, both surveys were designed to fulfill the basic assumption of maintaining standard procedures during trawl sampling (e.g. standard unit of time or distance) [88]. For this reason the surveys occurred in a consistent time period each year: late spring for "UWTV" (i.e. April, May) and autumn for "ScamPo" (i.e. September,

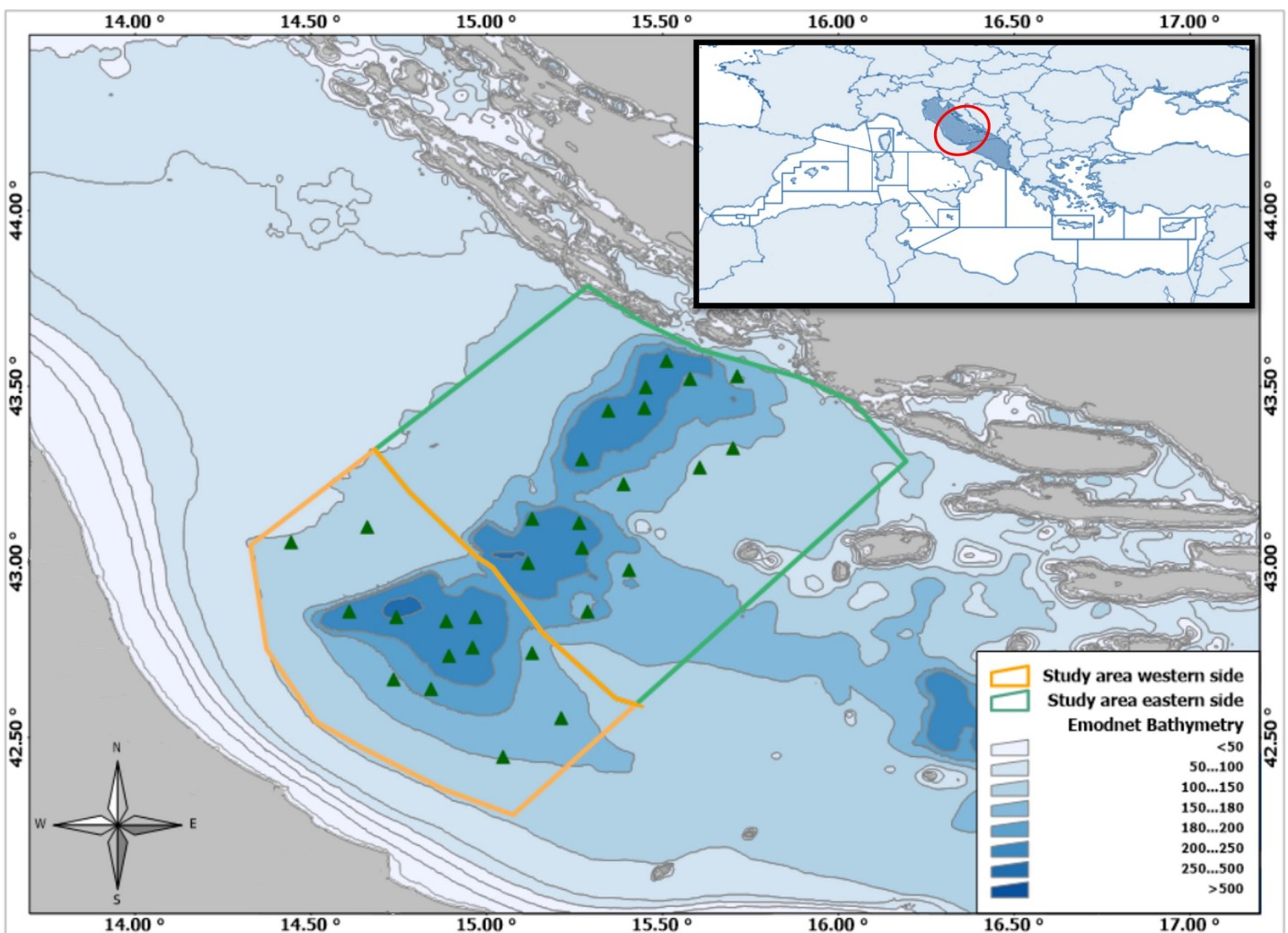

**Fig 1. The study area.** In the up-right rectangle the position of the study area within the Mediterranean basin is highlighted (red circle). The main map shows: the Pomo Pits Bathymetry (source: [75]), the boundaries of the study area, with western (orange polygon) and eastern (green polygon) sides divided by the Adriatic midline (source: [76]) and the trawl hauls planned for two considered surveys (green triangles).

October, November). The surveys were all carried out on board RV Dallaporta (LOA 35.30 m, 258 GT, 1100 HP). The 2 seasonal surveys share the same survey design (originally random stratified based on the bathymetric of 200 meters depth and following some legislative limitations) in the common area (i.e. western side) [72, 87]; some additional hauls have been added in both survey plans since 2016 (S2-S5 Figs in S1 File) [72]. Unfortunately, not all the planned hauls could be performed every year and in 2018 the entire spring survey was cancelled due to ship unavailability. However, the remaining performed hauls were equally allocated among areas with different management regimes. The trawling protocols are as well the same for both surveys [72, 87]: in accordance with the peak of *Nephrops* emergence [53, 60], hauls (1 hour duration) were all conducted between 100 m and 270 m depth at sunset and sunrise using an experimental net (mesh size of 22 mm in the body and 12 mm in the cod end) and trawl sensors to calculate the swept area [72]. In order to investigate possible discrepancies over time and space between indices trends, CPUE estimates for *Nephrops* for each haul were then calculated as total weight of the caught individuals divided by swept area (kg/km$^2$; hereafter referred to as

biomass index) and number of caught individuals divided by swept area (N/km$^2$; hereafter referred to as density index) (S2-S5 Figs in S1 File) [72]. In fact, for species such as *Nephrops* there could be possible differences due to the fact that density may be size dependent [89]. Furthermore, some stock assessment tools can be fed alternatively by both biomass and density indices (e.g. Stock Synthesis 3 and C++ algorithmic stock assessment laboratory [90, 91]).

Due to the oceanographic peculiarities of the study area, CTD (Conductivity Temperature Depth) casts were also carried out during the surveys in order to capture spatio-temporal variations of environmental parameters at small scales [92, 93]. Relevant oceanographic parameters (i.e. temperature, salinity and oxygen) measured as close as possible to the seabed (hereafter bottom data) where *Nephrops* lives were thus used as proxy to assess environmental variability over time and space. Therefore, to collect temperature and salinity data to be directly associated with each obtained CPUE value, CTD casts (by means of a *SBE19plusV2* probe) were performed as close as possible to the fishing location, before or after each haul [72]. The CTD was also equipped with a *SBE43* dissolved oxygen (concentration and saturation) sensor. A SBE5T pump ensures a constant flow over time inside the sensors.

## Model building

The effects of spatial and temporal variations of environmental parameters and fishery management measures on the CPUE indices were modeled by GAMs due to the high flexibility of the latter [31, 94]. Preliminary analyses (not reported here) provided confidence that merging the spring and autumn time series increased the model accuracy owing to a bigger sample size and thus reducing overfitting, as shown also in Wisz et al. [95]. However, the two time series were previously filtered only for observations including all the information to be tested in the model. Indeed it was not possible to include all the available hauls in the model building process because some of these were lacking values for some of the environmental parameters. All hauls carried out within the 2012 spring survey were excluded due to lack of values for bottom dissolved oxygen saturation. For the rest of the spring survey time series, some hauls for each year were removed (including the only two cases for which CPUE was equal to 0; S2 and S4 Figs in S1 File): 5 hauls (50%) for 2013, 2 hauls (13.3%) for 2014, 1 haul (12.5%) for 2015, 3 hauls (21.4%) for 2016 and 2017, and 2 hauls (18.2%) for 2019 were excluded. For the "autumn time series only one haul performed in 2017 was removed from the original dataset. A total of 56 (out of 89 available) hauls were selected within the spring time series from 2013 to 2019 (except for 2018), while in the autumn time series 36 (out of 37 available) hauls were selected.

The distribution of the response variables, i.e. biomass and density indices, appear not to be symmetrically distributed around the average with positively-skewed values and some outliers (e.g. highest values: 150 kg/km$^2$ for the biomass index and 17281 N/km$^2$ for the density index). Therefore, in the GAMs, a Gamma distribution with logit link function was assumed for both response variables; this distribution is one of the most used for environmental analysis [96], works well for positive-skewed data [97] and, owing to the logit link function, makes the model additive and therefore easier to interpret. To build the models, the covariates taken into account were: latitude (Y), longitude (X), depth (D), year (Yr), week of the year (week), time of the day (ToD), bottom temperature (BT), bottom salinity (Sal), bottom dissolved oxygen saturation (Oxy), and a factorial covariate accounting for restrictions on fishery (Fishery). The spatial scale is expressed by the "X" and "Y" covariates (in decimal degrees), which are the geographical coordinates referred to planned starting point of each performed haul, and by the "D" term (in meters) standing for the mean depth at which each haul was carried out. The spatial covariates are very important for *Nephrops* CPUE due to its sedentary behaviour [98]. The factor "Fishery" consists of 3 levels fitting with all the implemented managerial actions: fishery

allowed (Y), limited fishery (L) (i.e. buffer zones regulation), and fishery ban (N). The temporal covariate "Yr", expressed in numbers, was directly included as an interaction with the factor "Fishery" (using the "by" argument in the smooth functions) in order to model potentially different trends over years for each of the different fishery management levels. The inclusion of the intercept terms for categorical factor, as described in Wood [99], aims to increase the flexibility of the model fitting overall intercept differences between factor levels, and avoids artifacts; therefore, the "Fishery" factor was expressed both as intercept and as interaction with the covariate "Yr". Furthermore, the temporal scale was also expressed at different extents within the model equation (i.e. Year, week and ToD) in order to investigate the variability in catches between and within the seasons and times of day. Preliminary analyses (not presented here) carried out on the available dataset showed evidence that the "week" term, expressed as the number of the week within the year, is more suited (improves model performances) than other possible temporal scales (i.e. season or month during which the survey was conducted) to allow predictions by season (i.e. for the two survey seasons). The "ToD" covariate is instead a categorical factor representing the hauling time with two levels: sunset and sunrise (moments of the day at which correspond the peaks of maximum emergence from burrows of the species [55]). The terms accounting for the possible effects of environmental parameters recorded in correspondence of each trawl haul close to the seabed and already known to influence *Nephrops* life cycle and emergence behaviour (such as dissolved oxygen [40], salinity [50] and temperature [51], expressed as saturation percentage, practical salinity units, and *degree Celsius* respectively) were as well included in the models.

Before fitting the models on the data, in order to avoid multicollinearity, a covariate selection through Variance Inflation Factor (VIF) with backward selection was firstly carried out [100]. As a rule of thumb, the covariates with higher VIF should be excluded from the analysis one by one, until all the remaining variables have a VIF $< 3$ [101, 102]. The analysis showed a high multicollinearity value (VIF $> 3$) only for the longitude covariate; hence the "X" term (mainly correlated with the latitude "Y") was excluded from the considered models.

## Model selection

According to the backward selection suggested by Zuur [100], a full model (hereafter referred to as initial model or "modINITIAL") featuring all previously listed explanatory variables was formulated as follows for both density and biomass indices:

$$
\begin{aligned}
\text{CPUE} = \beta^0 + s_1(Y) + s_2(D) + s_3(BT) + s_4(Oxy) + s_5(Sal) + s_6(\text{week}) \\
+ s_7(Yr, by = \text{Fishery}) + f_1(\text{Fishery}) + f_2(\text{ToD}) + \varepsilon
\end{aligned}
\tag{1}
$$

Where CPUE corresponds to *Nephrops* biomass or density indices, $\beta^0$ is an overall intercept, $s_i$ are penalized cubic regression splines, $f_i$ indicate a categorical factor, the *by* symbol indicates that a spline function is separately estimated for each level of the factor, and $\varepsilon$ is the error term. The models were all settled with Gamma error distribution with a logit link function and REstricted Maximum Likelihood (REML) as smoothing parameter estimation method [102]. Based on preliminary analyses (not reported here) a limit of 6 was selected as the maximum number of basis functions (k) due to the low number of data.

In the aim to investigate how including environmental variables and fisheries management measures at the same time in the model would perform and influence the quality of the CPUE derived estimations, three simplified models were as well tested and compared with the initial: one model excluding from "modINITIAL" both the effects of environmental variables and fishery management actions ("modNOEM"), one excluding the effect of the considered environmental variables ("modNOE") and another excluding only the effect of fishery

management actions ("modNOM"). The three simplified models were as well applied both to biomass and density; "modNOEM" was formulated as follows:

$$CPUE = \beta^0 + s_1(Y) + s_2(D) + s_3(week) + s_4(Yr) + f_2(ToD) + \varepsilon \tag{2}$$

"modNOE" was formulated as follows:

$$CPUE = \beta^0 + s_1(Y) + s_2(D) + s_3(week) + s_4(Yr, by = Fishery) + \\ f_1(Fishery) + f_2(ToD) + \varepsilon \tag{3}$$

while "modNOM" was as follows:

$$CPUE = \beta^0 + s_1(Y) + s_2(D) + s_3(BT) + s_4(Oxy) + s_5(Sal) + \\ s_6(week) + s_7(Yr) + f_1(ToD) + \varepsilon \tag{4}$$

In order to evaluate which of these four models was the best, intended as the most informative, a comparison among the Akaike Information Criterion (AIC) values was conducted for both indices. Indeed, the AIC accounts both for the goodness of fit and the complexity of the model allowing to select the best according to the lowest value [103]. As recommended by Zuur, the expected result was to select the model featuring all explanatory variables introduced according to the known biology and behaviour of the species (i.e. initial model), otherwise the simplest model (fewest covariates) would have been the choice [100]. A 10 k-fold cross validation analysis was thus performed according to [104]. Therefore, for both CPUE datasets (biomass and density indices) and for each of the four equations, 10 runs were carried out, using a random defined 90% of the dataset to train a model. This operation was repeated 10 times, each time excluding a different portion of the dataset (corresponding to 10%). Then the performance of each trained model was evaluated using AIC. The obtained 100 AIC values for each of the models ("modINITIAL", "modNOEM", "modNOE" and "modNOM") were assessed for homogeneity of the variance by the Levene's test and then (in case of homoscedasticity) analysed by means of a parametric one-way ANOVA with post-hoc Tukey test [105]. The ANOVA was conducted to evaluate if there was a significant difference within the means of the AIC values across the four tested models (levels of the factor) for both indices.

The model selected by means of the above described AIC comparison was then further refined excluding the non-significant terms. The significance of each covariate was verified through an evaluation of the 100 p-values obtained from the cross-validation for both indices; as a rule of thumb, if in 80% of cases the p-values was above 0.05, thus the covariate was retained in the equation. After that, in order to evaluate the model prediction performance by the Root Mean Squared Error (RMSE), a second 10 k-fold cross validation analysis repeated 10 times was again run for both initial and selected model. Once verified that the latter model, including only the significant covariates, was the most reliable to conduct predictions on both the CPUEs, this was further refined tuning it on the entire dataset. The adaptation of the models to the data was evaluated by computing the proportion of the null deviance explained (i.e. percentage of deviance explained) and the adjusted $R^2$ [97]. Furthermore, the deviance explained by each term of the final model was estimated from the model outputs according to Wood et al. [99]. The final model was meant to be used for indices predictions to be carried out on the whole area of interest (see paragraph below). All the statistical analyses were carried out using the statistical software R ver. 3.5.2 [106], the associated "mgcv" and "car" packages [97, 107].

## Indices predictions

In order to carry out, through the final model, predictions of the CPUE indices over the study area, a spatial grid of the Pomo/Jabuka Pits area, 2x2 nautical miles, was built by means of the Geographic Information System (GIS) Manifold® System Release 8 (http://www. georeference.org/doc/manifold.htm). Each cell is identified by geographic coordinates (Lat = Y, Lon = X) corresponding to its center; average depth values were as well assigned to each cell by means of a source layer (available from: [75]) and the GIS Spatial Overlay function. Furthermore, the grid was replicated for each survey/year combination according to their spatial domain (i.e. autumn surveys are conducted only in the western side; Fig 1). The fishery management measures in place (according to the regulations in force during the various survey periods) were as well assigned to each cell. In order to inform each cell of the grids, at survey (as a proxy of season) and yearly levels, with average bottom values for each considered environmental parameter (i.e. BT, Oxy and Sal), it was taken advantage of the direct availability of reliable data sets; the same values collected during each survey and previously used in association to each recorded CPUE value to build the GAMs, were thus interpolated using a kriging method by means of the Surfer software (Surfer® 11.6 from Golden Software, LLC: www.goldensoftware.com). The obtained layers were then superimposed on the grids by means of the GIS in order to assign average bottom values to each single grid cell (using again the Spatial Overlay function; S6–S11 Figs in S1 File). Notwithstanding the dissolved oxygen percentage was not measured in spring 2012, the values recorded in the same season of the following year (i.e. 2013) were assumed as proxy within this study; a similar assumption was made for spring 2018 (missing year in the time spring series) for which all environmental values were simulated using those recorded in spring 2019. In fact, to pursue the aims of this study it was decided not to include datasets derived from different sources that could possibly introduce different estimation bias.

All predictions were then carried out by cell, also considering a selected level of the "ToD" term (sunrise) to standardize for the daytime temporal domain while at the same time modelling differences in CPUE within seasons/years. Since each cell had the same surface, predictions obtained for each cell/survey/year combination by means of the final GAMs (see paragraph above), for both biomass and density indices, were then averaged (according to the spatial domain of each survey) to obtain new standardized indices that were then compared with the mean observed CPUE (average of hauls values).

# Results

## Model selection and fitting

The Levene's test indicated homogeneity of the variance for the AIC values obtained from all the tested models ("modINITIAL", "modNOEM", "modNOE", "modNOM"), both for the biomass index (F = 0.2296; df = 3.396; p-value = 0.8757) and density index (F = 1.0907; df = 3.396; p-value = 0.352). The results of the parametric one-way ANOVA showed a significant difference in the means of the AIC values (calculated on 100 values through cross-validation) across the different tested models (grouping variable) both for biomass index (F = 501.7; p-value = 2e-16) and density index (F = 176.1; p-value = 2e-16; S1 and S2 Tables in S1 File). The Tukey post hoc test on the mean AIC showed significant differences for all the compared pairs of models both for biomass and density indices (S3 and S4 Tables in S1 File). "modINITIAL" has the statistically lowest average AIC value than "modNOE", "modNOM" and "modNOEM" (respectively not including environmental parameters, fishery management factors and both; S5 Table in S1 File); "modINITIAL" may thus be considered the best model. Indeed

"modINITIAL" resulted to be the most informative among the four tested GAMs, including at the same time the effect of environmental and fishery factors. The number of significant p-values obtained by the first cross validation analysis for all the covariates included in "modINITIAL" is shown in Table 1.

The "BT" covariate was seldom significant in "modINITIAL" both for biomass and density indices (respectively 18 and 9 times out of 100), while the "week" covariate was never significant (0 out of 100). Accordingly, these two covariates (i.e. "BT" and "week") were not included in the final models. Although the p-values obtained for the interaction between the covariate "Yr" and the factor "Fishery" were 0 and 1 (respectively for biomass and density indices) when the "Fishery" level was "L", this term was retained in both the final models because, for the two other levels of the factor ("Y" and "N"), the p-values were 100% of the times significant. Therefore, the final model ("mod FINAL") for both CPUE indices was formulated as follows:

$$CPUE = \beta^0 + s_1(Y) + s_2(D) + s_4(Oxy) + s_5(Sal) + s_7(Yr, by = Fishery) + f_1(Fishery) + f_2(ToD) + \varepsilon \quad (5)$$

The RMSE of the final model evaluated by running a second time the 10 k-fold cross validation analysis repeated 10 times resulted in lower averages than the initial model both for biomass and density indices (S6 Table in S1 File); it was thus considered as the one with the best prediction performance. The final model (Eq 5) included only significant explanatory terms; most of them were retained as single effects (i.e. Y, D, Oxy, Sal, Fishery, ToD) except for the interaction between "Yr" and "Fishery" factors. Tables 2 and 3 show the model parameters as tuned on the whole dataset. The final model explained 63.4% of the deviance (with an adjusted $R^2$ of 0.544) for the biomass index; while the explained deviance for the density index model was 66.9% (with an adjusted $R^2$ of 0.543). Percentages of deviance explained by each covariate of the final models both for biomass (kg/km$^2$) and density (N/km$^2$) indices are shown in the Supporting Information (S7 Table in S1 File). No specific issue emerged from the analysis of the residuals: they showed homogeneity of variance and a mean around zero (S11 and S12 Figs in S1 File).

Tables 2 and 3 also show the summary of the outputs of the final models for both the response variables. About the single categorical covariates, both tables show that the ToD term was significant and its effect was that average CPUE were lower at sunset than at sunrise

**Table 1. Number of significant p-values for each term of "mod INITIAL".** Number of significant p-values ($< 0.05$) out of 100 p-values in the 10 k-fold cross validation repeated 10 times for each covariate in the initial model for biomass and density indices.

| Covariate | Biomass index (kg/km$^2$) | Density index (N/km$^2$) |
|---|---|---|
| Y | 99 | 97 |
| D | 100 | 100 |
| BT | 18 | 9 |
| Oxy | 96 | 100 |
| Sal | 98 | 91 |
| Week | 0 | 0 |
| ToD | 100 | 100 |
| FisheryL | 95 | 99 |
| FisheryN | 89 | 57 |
| Yr:FisheryY | 100 | 100 |
| Yr: FisheryL | 0 | 1 |
| Yr: FisheryN | 100 | 99 |

**Table 2. Summary of the outputs of the final GAM for the biomass index (kg/km$^2$).** Explanatory variables included the single effect of factor Time of the Day (ToD) and Fishery, Year-Fishery interaction (Yr:Fishery) for each level of the factor ("Y" fishery allowed, "L" limited fishery, "N" fishery ban), longitude (Y), depth (D), percentage of dissolved oxygen (Oxy) and salinity (Sal); "SE", standard error; "edf", estimated degrees of freedom; "df", degrees of freedom.

| Parametric coefficients: | Estimate | SE | t-value | Significance level |
|---|---|---|---|---|
| Intercept | 3.29 | 0.10 | 31.93 | p<0.05 |
| f(ToD)sunset | -0.39 | 0.10 | -3.74 | p<0.05 |
| f(Fishery)L | 0.65 | 0.22 | 2.99 | p<0.05 |
| f(Fishery)N | 0.66 | 0.23 | 2.88 | p<0.05 |
| **Smooth terms:** | **edf** | **df** | **F** | |
| Yr: FisheryY | 0.96 | 5 | 5.29 | p<0.05 |
| Yr: FisheryL | 0.60 | 3 | 0.51 | 0.11 |
| Yr: FisheryN | 1.58 | 3 | 7.35 | p<0.05 |
| Y | 0.90 | 5 | 1.88 | p<0.05 |
| D | 3.88 | 5 | 16.15 | p<0.05 |
| Oxy | 1.57 | 5 | 1.66 | p<0.05 |
| Sal | 0.92 | 5 | 2.20 | p<0.05 |

(about 32% and 30% lower in case of biomass and density indices, respectively). Also the Fishery covariate had a significant impact and its effect was that with limited fishery (level "L") average CPUE were 92% and 58%, respectively for biomass and density indices, higher than when fishery was allowed; on the other hand, with no fishery (level "N") the average CPUE were 93% and 105%, respectively for biomass and density indices, higher than when fishery was allowed (Tables 2 and 3). The partial contribution of each continuous covariate for both biomass and density models are shown in Figs 2 and 3, respectively. The trend of Fishery interaction with the temporal variable (i.e. "Yr") (Figs 2 and 3, upper panels) showed a positive effect on catches when fishing effort was absent (level "N"), on the other hand a negative impact was highlighted in presence of an unmanaged fishing effort (level "Y"); no significant effect was detected with limited fishery (level "L"). As regard to the spatial terms, the spline of latitude (i.e. "Y") (Figs 2 and 3, middle left panel) had a slight influence on both models; the catch rates slightly decreased from the South to the North of the Pomo/Jabuka Pits area. The plots of both models indicated the "D" term (depth) as one of the covariates with the most

**Table 3. Summary of the outputs of the final GAM for the density index (N/km$^2$).** Explanatory variables included the single effect of factor Time of the Day (ToD) and Fishery, Year-Fishery interaction (Yr:Fishery) for each level of the factor ("Y" fishery allowed, "L" limited fishery, "N" fishery ban), longitude (Y), depth (D), percentage of dissolved oxygen (Oxy) and salinity (Sal); "SE", standard error; "edf", estimated degrees of freedom; "df", degrees of freedom.

| Parametric coefficients: | Estimate | SE | t-value | Significance level |
|---|---|---|---|---|
| Intercept | 7.61 | 0.11 | 72.43 | p<0.05 |
| f(ToD)sunset | -0.35 | 0.10 | -3.39 | p<0.05 |
| f(Fishery)L | 0.46 | 0.20 | 2.24 | p<0.05 |
| f(Fishery)N | 0.72 | 0.20 | 3.64 | p<0.05 |
| **Smooth terms:** | **edf** | **df** | **F** | |
| Yr: FisheryY | 1.71 | 5 | 9.34 | p<0.05 |
| Yr: FisheryL | 0.44 | 4 | 0.20 | 0.169 |
| Yr: FisheryN | 0.94 | 4 | 3.62 | p<0.05 |
| Y | 0.85 | 5 | 1.24 | p<0.05 |
| D | 4.02 | 5 | 22.21 | p<0.05 |
| Oxy | 2.05 | 5 | 3.31 | p<0.05 |
| Sal | 0.92 | 5 | 2.26 | p<0.05 |

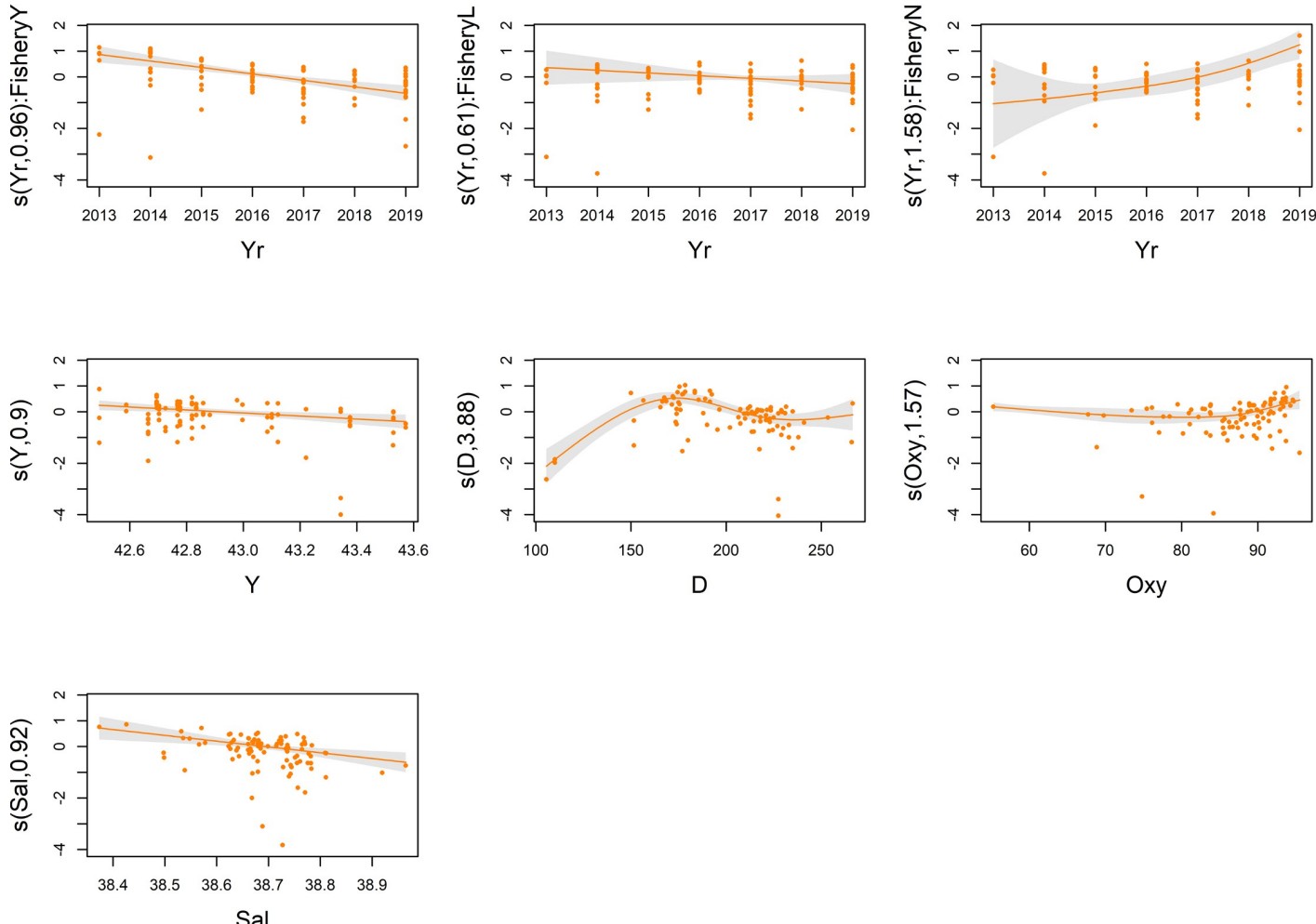

**Fig 2. Partial effects plots of GAM on *Nephrops* CPUE (kg/km²).** Partial effects (y axis) of spatial (Y, D), environmental (Oxy, Sal), and fishery management variables (Yr:Fishery) selected for the final GAM. Grey shaded regions indicate the 95% confidence intervals, dots are the residuals.

evident impact (Figs 2 and 3) showing a positive effect up to about 175 m, then the CPUE indices decreased since almost 225 m after which they become constant (Figs 2 and 3, central panel). The effect of "Oxy" for both CPUE indices (Figs 2 and 3, middle right panel) indicated a slightly negative impact on catch rates until almost 85% after which it becomes strongly positive, leading to a consequent increase in the CPUE indices. The effect of the "Sal" term (Figs 2 and 3, bottom left panel) showed a negative impact on catch rates when salinity values increase; these effects were observed for both models.

## Predictive distribution maps

Figs 4–7 show the annual predictions of *Nephrops* biomass and density CPUEs per cell, within the spatial domains previously defined by means of grids (in the Pomo/Jabuka Pits study area), for the spring and autumn time series.

Maps of the related standard errors (per cell) are available in the Supplementary Material (S14–S17 Figs in S1 File). Overall, the predicted distribution in space and time of the biomass CPUE (kg/km²) seems to follow the same patterns of the density one (N/km²), even if the

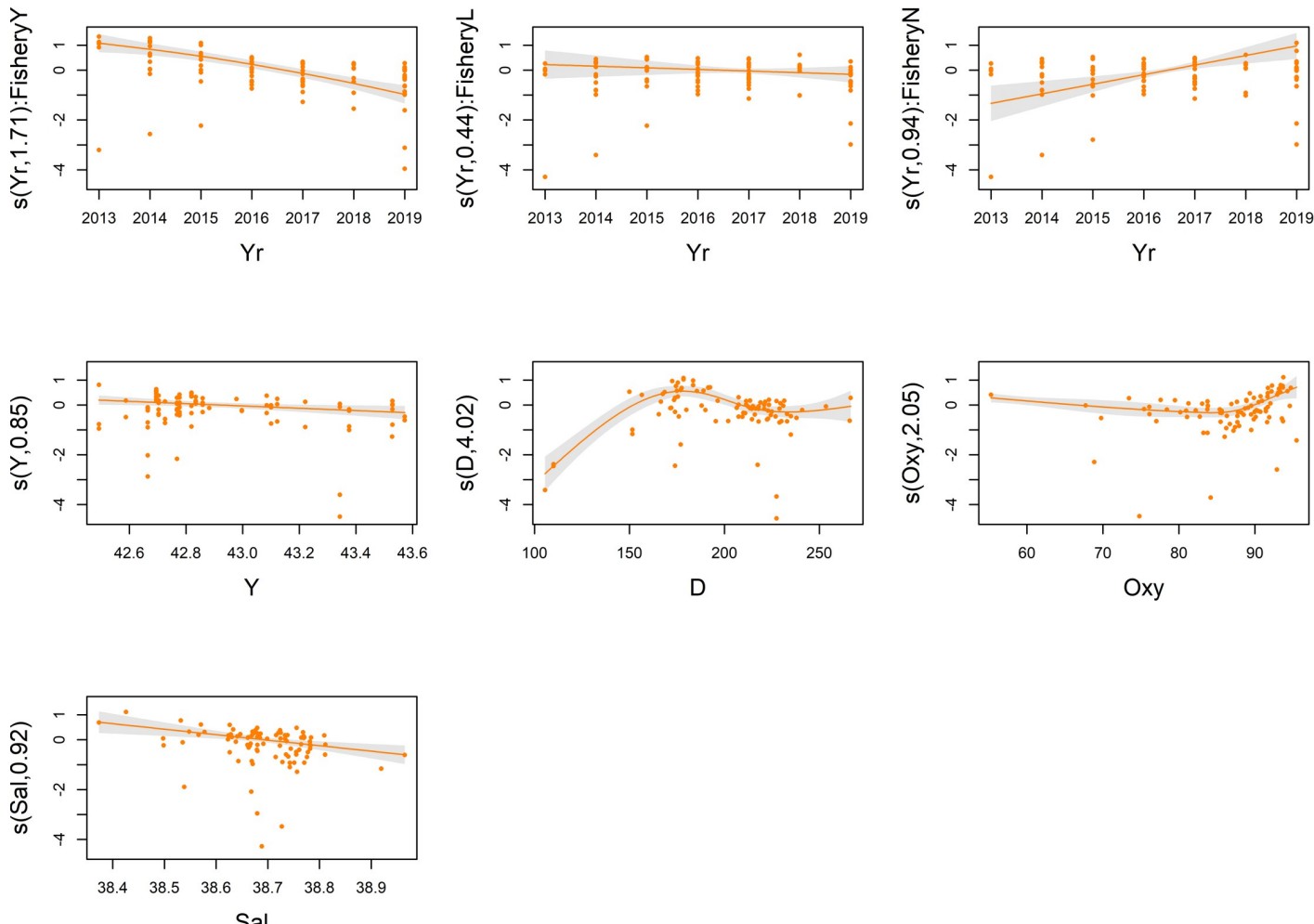

**Fig 3. Partial effects plots of GAM on *Nephrops* CPUE (N/km²).** Partial effects (y axis) of spatial (Y, D), environmental (Oxy, Sal), and fishery management variables (Yr:Fishery) selected for the final GAM. Grey shaded regions indicate the 95% confidence intervals, dots are residuals.

numerical predictions are of course different. The highest estimates for the spring time series in the period 2012–2017, were obtained for the cells localised around the Pomo/Jabuka Pits at depths from about 150 m to 200 m and in the south-eastern part of the study area. By contrast the distribution of *Nephrops* indices in the last two modelled years (i.e. 2018 and 2019) was almost pooled within the boundaries of the fishing ban zone (i.e. zone "A"). For the autumn time series, the highest predictions were centered in the southern part of the study area as well, with the exception of 2017, 2018 and 2019 for which the highest predicted catches were within the boundaries of the FRA zone "B", from about 180 m to 200 m of depth. All the modelled indices showed very low or even no predicted values for the cells in the north-west zone of the study area.

## Standardized CPUE values

Fig 8 shows a comparison between mean observed CPUEs (average of hauls values per year and survey) and mean values obtained by means of the final GAMs (average of values obtained for each cell/survey/year combination), for both biomass and density indices.

For the biomass models, the maximum difference between observed and predicted mean values in the spring survey time series was found in 2012 with an increment of about 65%

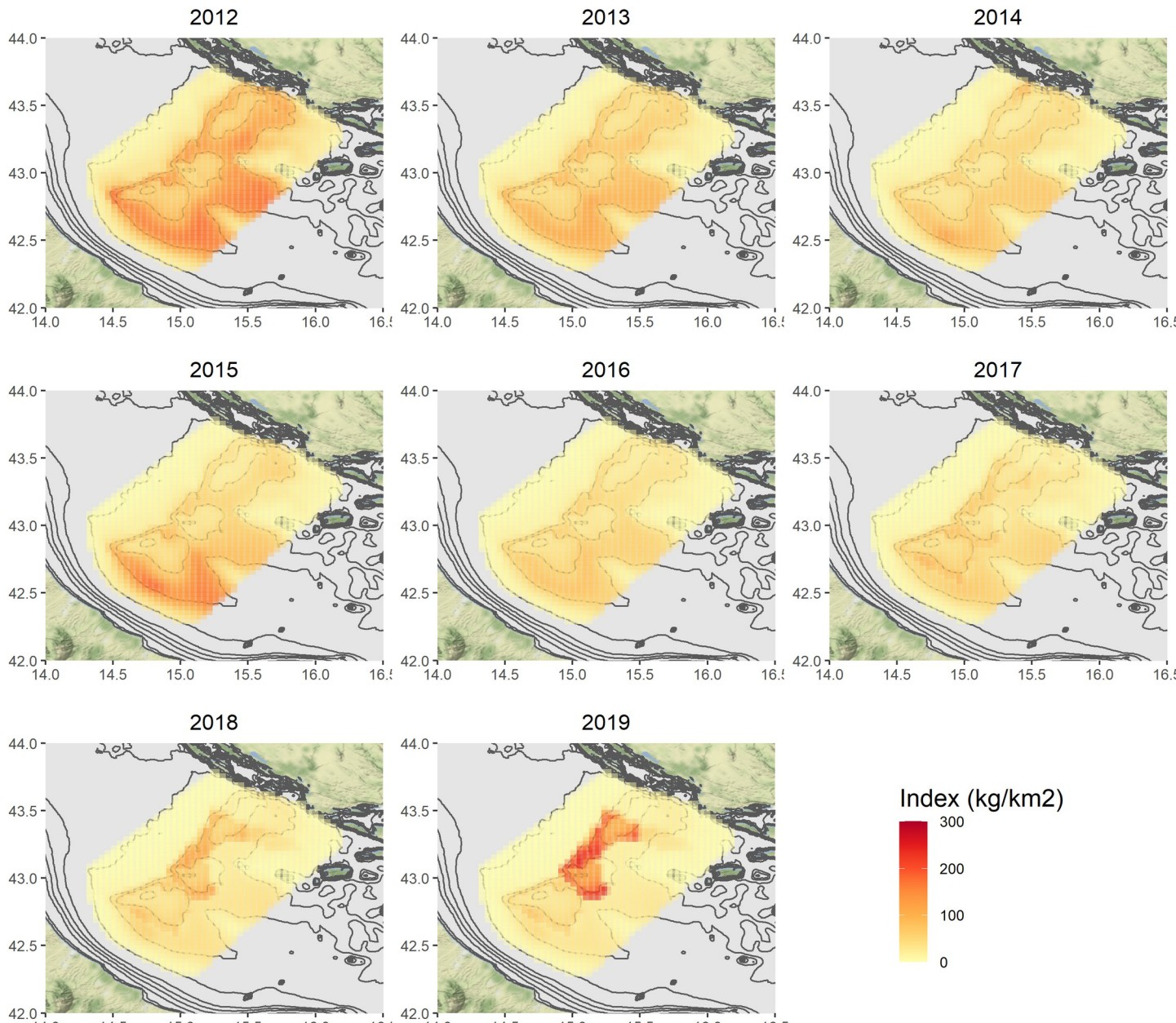

**Fig 4. Predicted spatial distributions of *Nephrops* biomass index (kg/km$^2$) for the spring time series.** Maps were made using the ggmap package [108] for R. Bathymetry layer source: [75]. Map tiles by Stamen Design, under CC BY Data by OpenStreetMap, under ODbL.

from 24.6 kg/km$^2$ to 71.3 (± 19) kg/km$^2$ (Fig 8, panels a); in the autumn survey time series an important difference was found in 2015 with an increment of about 33% from 24.9 kg/km$^2$ to 37.33 (± 9.1) kg/km$^2$ (Fig 8, panels b). Apart from these two exceptions, the mean predicted values for the biomass index were always within the interquartile range (i.e. middle 50% of the observed data) (Fig 8, panels a and b). For the density models, the highest discrepancy in the spring survey time series was as well detected in 2012, with a huge increment from about 2704 N/km$^2$ to 5723.27 (± 1910.1) N/km$^2$ (Fig 8, panel c). Differently from the biomass models, for density indices in the autumn survey time series the maximum differences of about 50% were

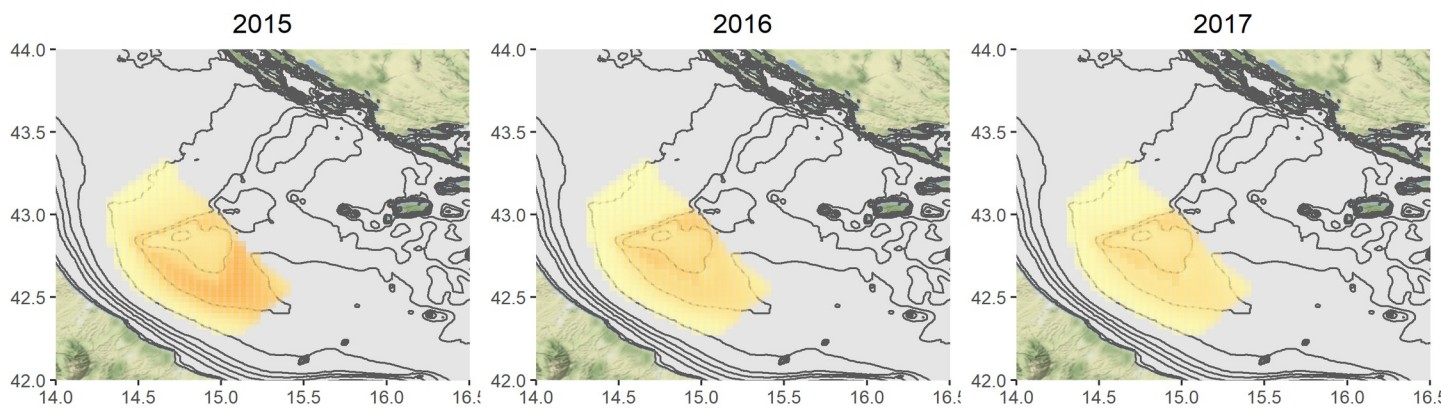

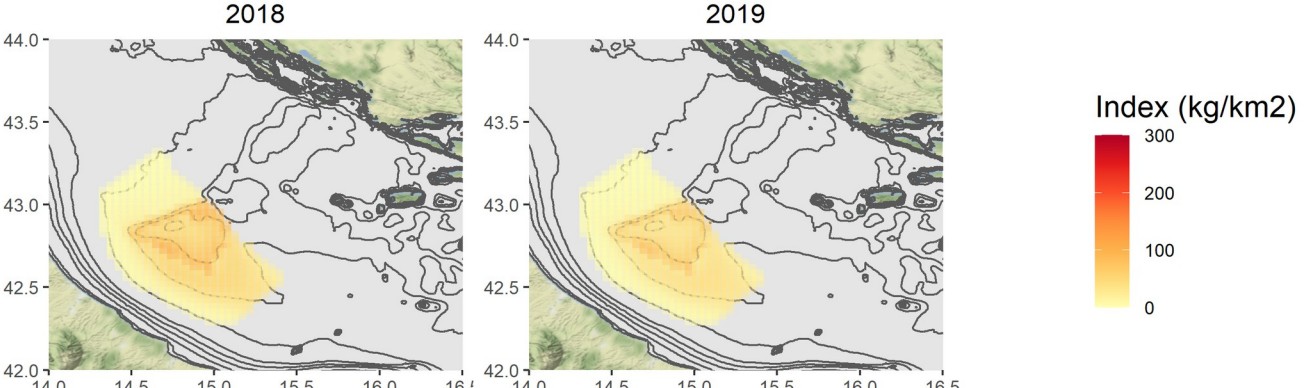

**Fig 5. Predicted spatial distributions of *Nephrops* biomass index (kg/km²) for the autumn time series.** Maps were made using the ggmap package [108] for R. Bathymetry layer source: [75]. Map tiles by Stamen Design, under CC BY Data by OpenStreetMap, under ODbL.

found in 2015 and 2017, with an increment from 2129 N/km² to 3263.3 (± 773.2) N/km² and a decrement from 2188 N/km² to 1134.9 (± 237.3) N/km² respectively (Fig 8, panel d). Excluding these cases, all the mean predicted values were always within the interquartile range of the observed data (Fig 8, panels c and d). With the exception of 2018, the mean values of the predicted CPUE for the spring survey time series are higher than those for the autumn survey time series (Fig 8). A concordance between the observed CPUE trends and the predicted ones was confirmed for the majority of the years within both time series. The mean predicted spring CPUE trends showed the highest peaks for both abundance and density indices in 2012 and 2015, together with the highest mean standard errors, and a slight increase of estimates in 2019 (Fig 8A and 8C). For the mean predicted autumn CPUE trends, the highest peaks and standard errors for both indices were detected in 2015 and in 2018 (Fig 8B and 8D). In general, for both the spring and autumn time series, biomass and density trends seem to agree over time, apart from a slight difference between 2018 and 2019, when biomass slowly increases (Fig 8A and 8B) while density remains stable in spring (Fig 8C) and weakly decreases in autumn (Fig 8D).

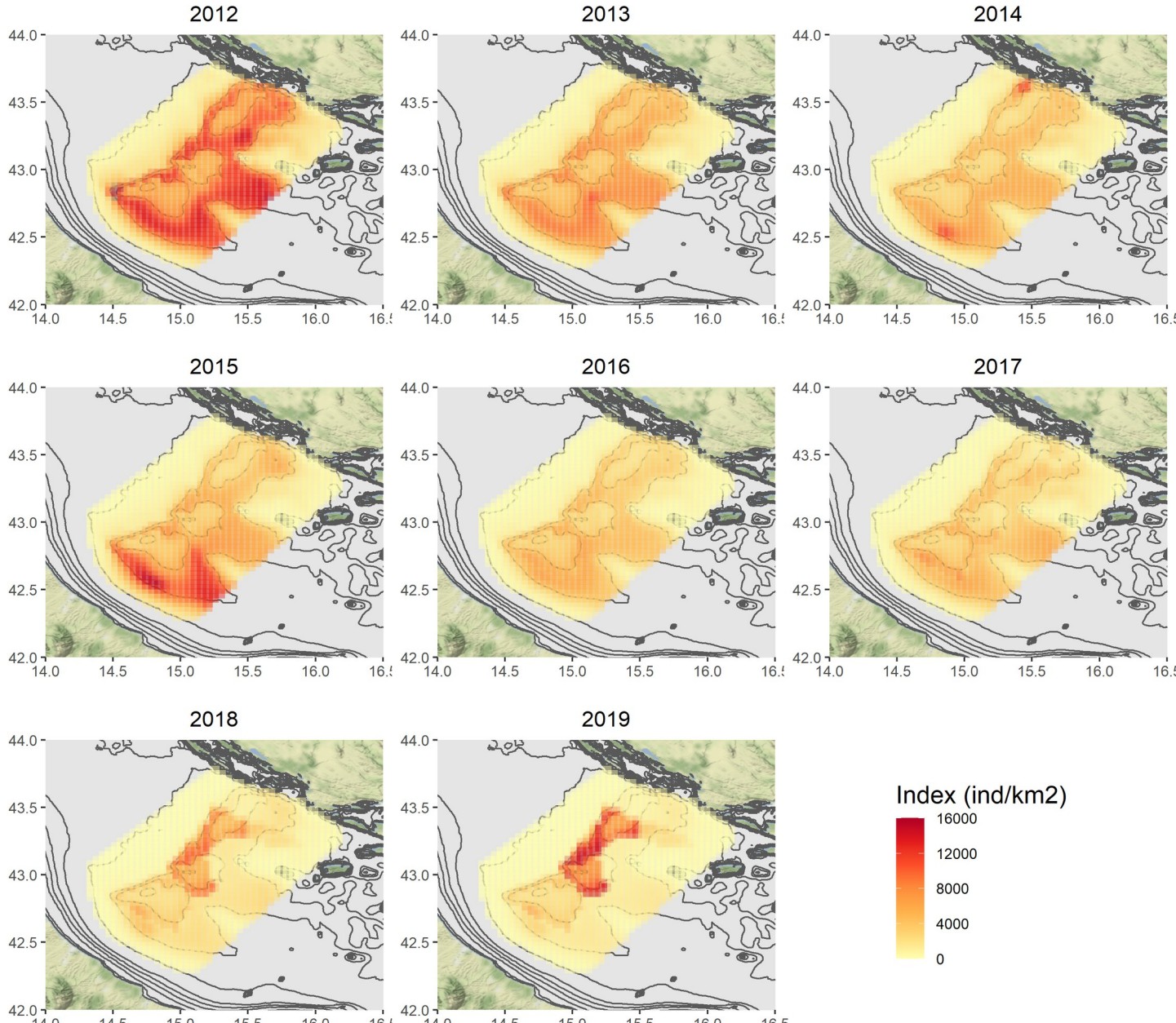

**Fig 6. Predicted spatial distributions of *Nephrops* density index (N/km²) for the spring time series.** Maps were made using the ggmap package [108] for R. Bathymetry layer source: [75]. Map tiles by Stamen Design, under CC BY Data by OpenStreetMap, under ODbL.

## Discussion

In this study, GAMs were applied to produce specific standardized *Nephrops* indices accounting for both the effects of environmental variables and fishery management actions in place within the Pomo/Jabuka Pits (central Adriatic Sea). The raw CPUE time series treated in this work were never included in an official stock assessment before, thus the information here provided could also be relevant to feed future robust population dynamics models for this species. The slight difference between the trend of biomass and that of density from 2018 to 2019 may indicate a possible size effect (i.e. larger individuals leading to higher biomass and lower

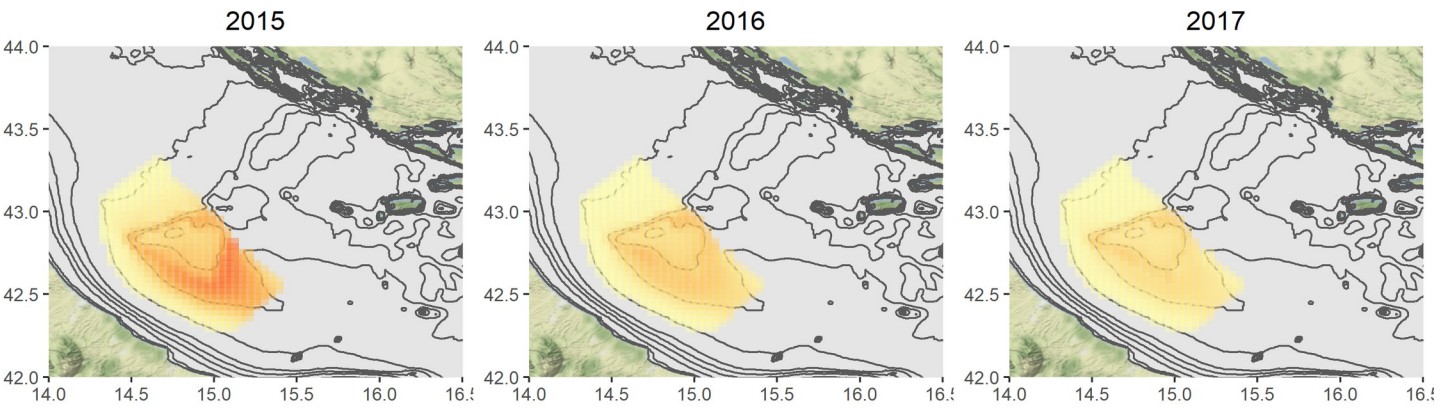

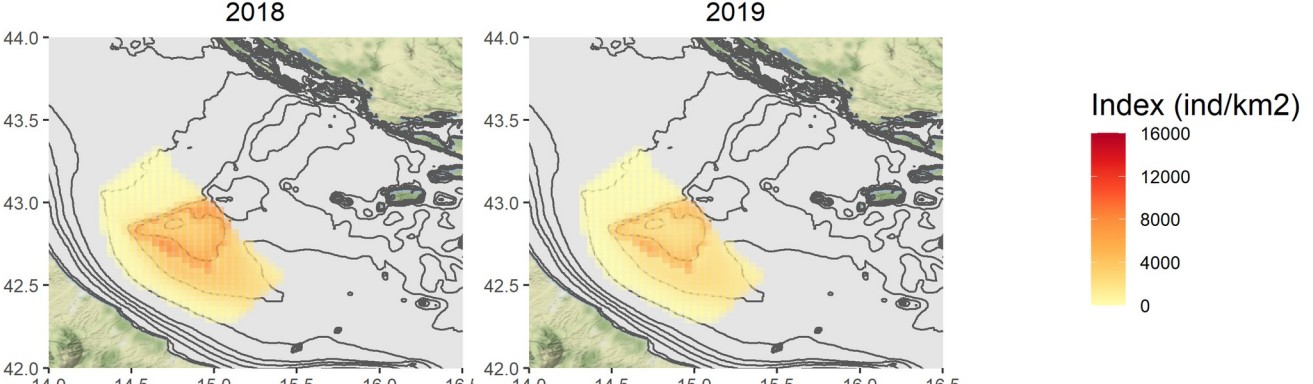

**Fig 7. Predicted spatial distributions of *Nephrops* density index (N/km²) for the autumn time series.** Maps were made using the ggmap package [108] for R. Bathymetry layer source: [75]. Map tiles by Stamen Design, under CC BY Data by OpenStreetMap, under ODbL.

density), possibly induced by the latest management regime; further studies on the size structure of the population would therefore be worth noting.

## Filling the gaps of a time series through standardization with GAMs

Poor weather conditions, limited shiptime, unavailability of vessels, equipment failures and other occasional issues could lead to shortcomings due to data unequally distributed in space and time along the survey time-series [88]. It is well known that in case of fishery-dependent data, sampling strategies inadequate to the extension of the study area may result in ostensibly stable catch rates (i.e. hyperstability) or in not reliable trends changes (e.g. hyperdepletion) [109]. In case of fishery-independent surveys, spatial stratification could be adopted to minimize such bias in the abundance estimates so long as each stratum has an approximately homogeneous density [110]. Moreover, when the study area is not uniformly surveyed due to biased sampling design (e.g. an occasional lack of data in some planned sampling locations) standardization of catch rates is recommended to remove most of the variability not directly

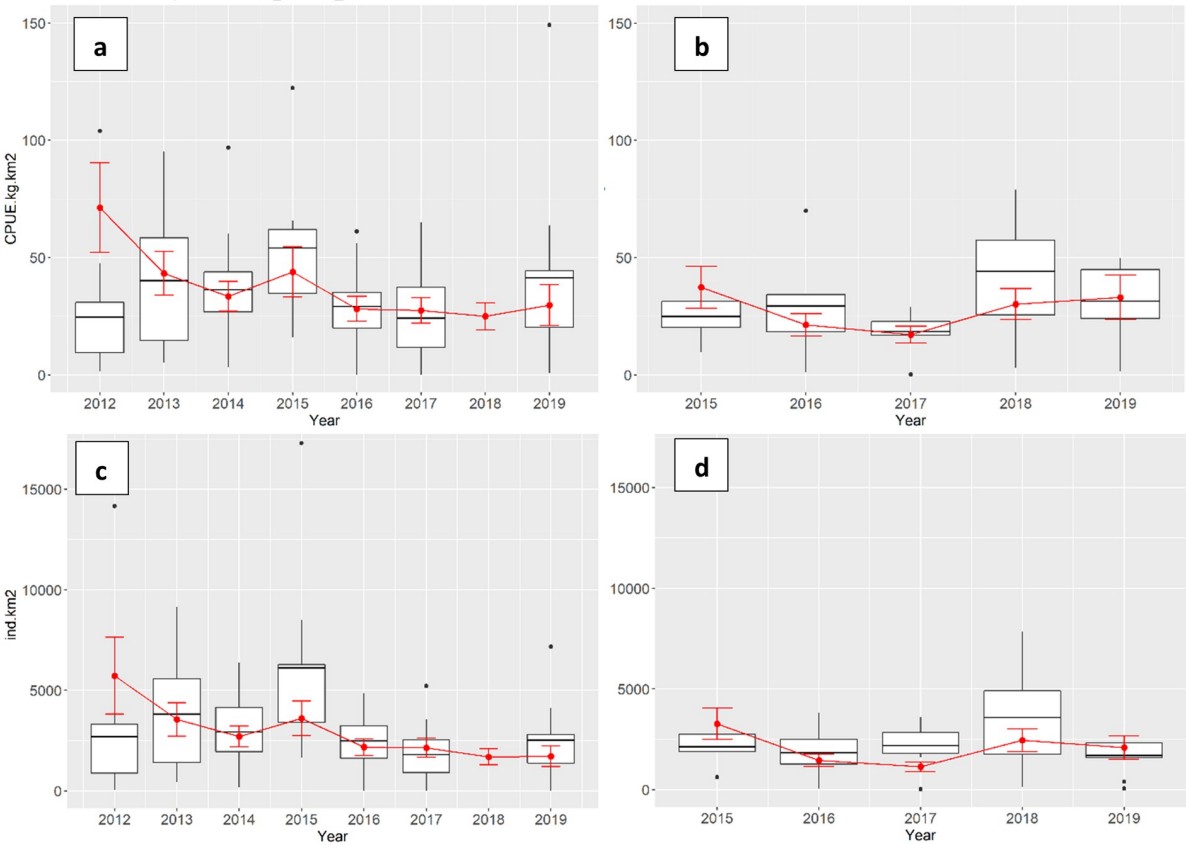

**Fig 8. CPUE indices.** Boxplots for the unstandardized time series over years represented with mean values (horizontal black lines), third and first quartiles (top and bottom vertical black lines, respectively) and outliers (black dots) plotted against the mean of predicted annual CPUE obtained for each cell of the grid (red dots) with standard errors (red lines). The spring time series is on the left (*a* and *c* panels), the autumn time series is on the right (*b* and *d* panels). Biomass index is represented in the upper panels (*a* and *b* panels), while density index in the lower panels (*c* and *d* panels).

attributable to changes in abundance [30]. Distribution models including environmental effects should thus be preferred to predict species distributions and abundances at local scales [111, 112]. However, when dealing with gaps in the spatiotemporal distribution of marine resources GAMs are more convenient than GLMs because they can easily incorporate the non-linear responses of catches to geographic factors by smoothing rather than stratifying [32, 33]. On the other hand, GAMs are likely to cause overfitting, especially with small sample sizes, because they allow the use of several fixed effects in nonlinear smoothing functions, which often reduce the predictive ability [113]. In this study, GAMs were selected as a suitable tool for filling the gaps inside the time series and at the same time explain the influence of environmental processes on the species distribution. In the future, it would be interesting to compare this standardization modelling approach with other spatio-temporal models which also include autoregressive processes [35, 114].

## Spatio-temporal distribution of estimated CPUE

Given that the seasonal emergence rhythmicity of *Nephrops* is historically demonstrated in the Adriatic Sea [57], an intra-annual temporal term (i.e. "week") was thus included in the model building process: unfortunately no significant differences between catch rates from the 2

surveys were found. Moreover, the available data set was probably too little extended over the years to allow the detection of intra-annual CPUE variability. However the informed grids allowed to predict seasonal indices relying on the availability of environmental data for spring and autumn. Seasonal differences in emergence patterns were addressed for different stock of a similar species, *Metanephrops challengeri* (Balss, 1914), by using assessment models able to include seasonal abundance indices [115]. Therefore, it would be advisable to develop in the near future similar solutions also for Mediterranean *Nephrops* stock, as suggested by Aguzzi et al. [51]. Hence, it would be important to maintain the consistency of the 2 *Nephrops* targeted seasonal time series in order to improve the understanding of spatio-temporal dynamics and to provide increasingly accurate stock assessment inputs. Furthermore, when dealing with small data sets it is not possible to select a large number of predictors, otherwise this could lead to an increase in the uncertainties [99]. Hence, in the model building process, only the main effects for all the covariates were tested with the exception of the managerial factor which was expressed both as intercept and as interaction with the covariate "Yr". As recommended by Wood, an intercept term needs to be included when using ordered factors because smooths with factors are affected by the centering constraints [99]. As might be expected, both GAMs estimate increasing catch rates when fishery is absent (level "N") and decreasing ones when fishery is allowed (level "Y"); on the other hand it is interesting to note that in presence of partial fishery restriction measures (level "L") the catches do not increase but remain stable. This could be explained by: (i) a time-series not long enough to observe significant changes in the abundance; (ii) a too little reduction of fishing effort in order to observe an increase in the catch rates. However, following the implemented regulations and their changes in time and space, the levels of the "Fishery" factor for fishing ban ("N") or limitations ("L") were fixed in the grid from autumn 2015 forward; given the actual explanatory power of "Fishery" factor, it is therefore plausible that since then the managerial actions influenced the distribution of the estimated indices. This is also supported by some recent studies showing a clear alteration of size and density in crustacean species when a no-take-zone or fishery ban periods are established [116–118]. Indeed, considering the predictions for the cells set at level "N" for the Fishery factor from 2018 onwards (corresponding to the fishery ban area "A") an increment in CPUEs was found. Furthermore, the highest values of catches are mainly concentrated in the cells positioned around the Pits at a bathymetric range from about 180 m to 200 m. Actually, the depth variable proved to be one of the main factors influencing the spatial distribution of *Nephrops*. Both models presented here show that there is a positive effect on the catch rates for the above mentioned bathymetric range, suggesting the presence of higher values of density and biomass indices along the borders of the Pits rather than within the Pits (< 200 m).

Overall, a slightly positive spatial gradient from the North to the South of the Pomo/Jabuka Pits area is supported by the spline of the "Y" term. Such an effect can be easily observed in the predictions for the period before the implementation of the first management measure (i.e. spring 2015) for which the spatial distribution of the indices within the maps were mostly centered in the south-east zone of Pomo/Jabuka Pits. A pretty similar distribution of Landings Per Unit of Effort (LPUEs) was previously estimated by a model developed by Russo et al. [56] which estimates *Nephrops* landings per fishing ground combining landings per harbor and fishing vessels routes from VMS (Vessel Monitoring Systems [119]). Besides, a displacement model by Bastardie et al. showed a possible redistribution of the fishing effort toward surrounding areas in case of establishment of a fishing ban within the Pomo Pits [28]. This could therefore support the reduction of catches in the southern part of the prediction area (not subject to fishery limitations) after the implementation of the management measures. The agreement with the local (i.e. Pomo/Jabuka Pits) spatio-temporal distribution of *Nephrops* is an evidence of the great spatial accuracy of the models presented here.

## Roles of environmental variables in the standardization models

Changes in biomass and density over time and space are the results of interactions among individuals and between individuals and their environment; hence, understanding of adaptive behaviours in response to changes in the environment helps to explain the complexity in spatiotemporal distribution [109]. One of the main processes influencing the environmental conditions in the Pomo/Jabuka Pits is the periodical, though occasional, water mass renewal caused by dense water formation over the northern Adriatic shelf [73, 120, 121]. The Middle Adriatic Deep Water (MAdDW), that resides throughout the year in the bottom layer of the Pomo/Jabuka Pits, is periodically renewed by Northern Adriatic Densev Water (NAdDW) at one to three years intervals but occasionally at longer intervals [73, 120], leading to a density and oxygen increase and to a temperature and nutrient decrease [122]. Such renewals have an effect on the local biodiversity and on the trophic status of benthic communities [123].

In the model selection section, the initial model was built and then compared to other three models derived firstly by removing both the environmental covariates and the terms related to the fishery management actions, secondly by removing only the environmental covariates and finally by removing only the fishery management terms. The aim was to statistically understand if the environmental and fishery covariates could effectively improve the model performances. The results of statistical analysis provided evidence that the model including both environmental and fishery covariates perform better than the other three. In both final models, a positive impact on catch rates was estimated for dissolved oxygen values greater than 85%. An exceptional dense water formation was reported during winter 2012 in the Central Adriatic Sea [124, 125]. An average bottom dissolved oxygen saturation of 91% was recorded in late March 2012 in a transect carried out in the western side of the Pomo/Jabuka Pits area, while an average of 81% was recorded for the same stations in April 2013 (CNR IRBIM unpublished data); thus the use of the 2013 dissolved oxygen values as a proxy for 2012 could probably have led to underestimation of the modeled indices in 2012 if not coupled with other explanatory environmental variables (i.e. salinity; see below). On the other hand, relatively high catches were estimated also for dissolved oxygen values lower than 60%. Laboratory studies show that low percentages of dissolved oxygen saturation within the substratum may force *Nephrops* individuals out of their burrows in an attempt to ventilate on the bottom surface [126]. In the Pomo/Jabuka Pits the bottom oxygen depletion caused by local "ageing" consequent on reduced ventilation could even lead to sediment hypoxia (less than 40% [122]). Historically, extraordinary hypoxic conditions in the Central Adriatic Sea associated with a sharp decline in both *Nephrops* landings and experimental catches were reported during the 1980's [127]. Such a stressful environmental condition could influence the catchability and the natural mortality of this species [40, 128, 129] even affecting the strength of recruitment [130]. Similarly, eutrophication events affecting the Kattegat and Skaggerak fishery in the mid-1980s led to a steep decrease of catch rates, as a consequence of high mortality [131, 132]. No values attributable to hypoxic conditions were recorded along the time series presented here. However, very low levels of dissolved oxygen saturation were recorded in the area before the exceptional dense water formation in winter 2012 (e.g. 55–52% in 2010 and about 47% in 2011; CNR IRBIM unpublished data). Raw catch rates showed an increase of mean indices from 2012 to 2013, by contrast the model estimated a decreasing trend while maintaining high standard errors. Actually, the discordance between mean observed CPUE and the averaged prediction in 2012 for both indices could potentially also be influenced by the low level of oxygen saturation in the previous years which could have already reduced the abundance of *Nephrops*, similarly to what happened during 80's [127]. Given the fact that key environmental stressors could affect *Nephrops* at different life stages [133], future models should also be tested for the dissolved oxygen

saturation recorded in previous years in correspondence with recruitment times (in Adriatic hatching happens yearly in late winter [40]), as suggested for sea surface temperature and pink shrimp (*Parapenaeus longirostris*) in Colloca et al. [134].

Nevertheless oxygen is not the only environmental parameter that could affect *Nephrops* catch rates; salinity stress conditions as hypercapnia and salinity fluctuations may as well alter abundance of recruits [135]. However, literature on salinity influence on *Nephrops* is poor and mainly focusing on survival studies, generally showing an increase in mortality in low salinity ranges (around almost 15 in Harris and Ulmestrand [50]; and from about 24 to 29 in Fox et al. [136]). Within this study, in both models slight variations in salinity ranges resulted in relevant impacts on CPUE values, despite the measured range being really narrow. Both in spring 2012 and autumn 2015 (the periods in which the maximum CPUE predicted mean values were obtained for the two surveys) salinity showed generally lower estimates for the whole study area extent, if compared with those related to the respective following year/season combinations (S6 and S7 Figs in S1 File), suggesting that such high estimates in the prediction could be influenced by these low salinity concentrations. However, the reasons behind these discrepancies in predictions, associated with high standard errors, could also be due to an insufficient amount of collected data, in particular for 2012. On the other hand, a negative impact of high salinity levels as the one observed within this study has never been reported in literature, thus further studies on salinity variation effects on *Nephrops* populations are suggested. As stated before, in this work it was not possible to test more than one interaction at a time; it is thus recommended that future studies will also take into account interactions between different environmental variables during the model building process. Further model developments should also include and investigate the role of other parameters known to potentially affect the CPUE of *Nephrops* such as sediment composition, tide level and the lunar cycle [45, 48, 137].

## The importance of targeted sampling protocol for *Nephrops*

A common objective of population dynamics models is to employ observed CPUE indices to estimate the population abundance through time [109]. According to Cook [138], data collected by means of trawl surveys are a more accurate source of information for estimating stock abundance than observations from commercial landings. Despite that, owing to the temporal variation in *Nephrops* emergence behaviour, catch data from all survey activities may not be representative of the real population [55, 85]. For example, the MEDiterranean International Trawl Survey (MEDITS [139]) which is the main index of abundance available for Mediterranean demersal resources, is not properly designed to catch *Nephrops* (e.g. hauls start one hour after dawn and stop one hour before dusk every day), indeed its estimates are also affected by its rate of emergence as for commercial trawl fishery [140, 141]. In order to overcome these kinds of issues, in several European countries a specific methodology (namely UWTV) based on the use of a camera system for the detection of burrows numbers and derivation of density indexes was developed [142]. In trawl surveys to reach an adequate level of proportionality a representative portion of the population have to be sampled and the efficiency of the net have to be experimentally measured [143]. Both conditions are easily achievable during a fishery-independent survey thanks to a well-known and standard procedure [88, 144]. It is the case of the 2 Adriatic trawl surveys reported here which were specifically designed to target *Nephrops*; indeed the trawl hauls are all conducted at the peak of emergence of the species (i.e. sunset and sunrise [55]) and the experimental net was designed to catch a representative portion of the population including juvenile individuals [72, 87]. The reliability of the collected data is also proved by the almost total absence of zeros in the time series; such a condition is rarely confirmed in other demersal trawl surveys within the Adriatic Sea (e.g. MEDITS survey

[145]). Furthermore, the presence from 2015 of another *Nephrops* targeted survey carried out during autumn season (i.e. "ScamPo"; [72]) allowed to collect information about population structure at a finer timescale. Such a level of information is an important tool for stock assessment purposes as it allows to consider the different availability of *Nephrops* sexes during the year [40, 72, 146] improving the input quality of population dynamics models.

## Conclusions

The overarching goals of this study were to analyze how drivers such as environmental changes over time and alterations of fishing effort influence CPUE estimates and to use these outcomes to standardize the abundance indices from fishery-independent trawl surveys. The *Nephrops* subpopulation inhabiting the Pomo/Jabuka Pits was selected as a case study mainly due to behavioural characteristics of this species that may affect catches, the oceanographic peculiarities of the study area and the implementation and subsequent change in time and space of different fisheries management measures. The availability of trawl survey data collected along with environmental parameters allows to include in the model environmental effects directly related to the capture event. Among the investigated covariates the most explanatory ones resulted to be fishery and depth, while dissolved oxygen and salinity showed a relevant effect on catch rates. The standardization of CPUE from the targeted surveys allows predicting reliable indices for a sedentary species with a high variability in catch rate as *Nephrops*. In addition, through the use of GAMs, it was possible to fill the gaps within the historical time series, thus being able to provide a biomass or density index for all the considered time frame. Indeed, an informed grid was built in order to predict the indices at different spatial scales. The standardization for the two different time series led to estimate CPUE indices for each of the two seasons (i.e. spring and autumn) allowing to model the differences in the availability of *Nephrops* during the solar year. This is of relevant importance from the management point of view and, in future analyses, this standardized CPUE might improve the evaluation of the status of the resource at different temporal and spatial scales. Moreover, the achieved knowledge on the impact of fisheries and some environmental factors on *Nephrops* communities and their spatio-temporal dynamics in Pomo/Jabuka Pits could be of extreme relevance while developing an ecosystem approach to fishery management.

## Supporting information

**S1 File. S1-S7 Tables showing results of the parametric one-way ANOVA, Tukey multiple comparisons among AIC values, the mean AIC and RMSE values, and the percentages of deviance explained by each term of "mod FINAL" for both biomass (kg/km$^2$) and density (N/km$^2$) models. S1**-**S5 Figs** showing maps about the management measures implemented within the study area since 2015, bubble plots by year of normalized biomass and density CPUE for each haul of both spring and autumn surveys. **S6-S11 Figs** showing environmental maps of bottom salinity, bottom dissolved oxygen and bottom temperature by year for both spring and autumn time series. **S12-S17 Figs** showing plot about the residual analysis for the final model for both biomass and density indices together with prediction maps of the related standard errors for both time series.
(DOCX)

## Acknowledgments

This work represents partial fulfilment of the requirements for Matteo Chiarini's doctoral project carried out in the framework of the International PhD Program "Innovative

Technologies and Sustainable Use of Mediterranean Sea Fishery and Biological Resources (www.FishMed-PhD.org). The authors would like to thank Carlo Froglia for sharing his relevant and peerless knowledge about the study area and the target species, Jacopo Aguzzi for its helpful hints, Giuseppe Caccamo, Roberto Cacciamani, Federico Calì, Camilla Croci, Giordano Giuliani, Paolo Scarpini and Lorenzo Zacchetti as members of the scientific staff that carried out the surveys and the crew of RV Dallaporta for the precious help during the fishing operations. The authors also wish to thank the anonymous reviewers for their constructive comments that helped to improve the manuscript.

## Author Contributions

**Conceptualization:** Matteo Chiarini, Ian D. Tuck, Michela Martinelli.

**Data curation:** Matteo Chiarini, Michela Martinelli.

**Formal analysis:** Matteo Chiarini, Stefano Guicciardi, Federica Grilli, Pierluigi Penna.

**Funding acquisition:** Michela Martinelli.

**Investigation:** Matteo Chiarini, Filippo Domenichetti, Giovanni Canduci, Andrea Belardinelli, Damir Medvešek.

**Methodology:** Matteo Chiarini, Stefano Guicciardi.

**Project administration:** Michela Martinelli.

**Supervision:** Michela Martinelli.

**Visualization:** Matteo Chiarini, Michela Martinelli.

**Writing – original draft:** Matteo Chiarini.

**Writing – review & editing:** Matteo Chiarini, Stefano Guicciardi, Silvia Angelini, Federica Grilli, Pierluigi Penna, Alberto Santojanni, Enrico Arneri, Nicoletta Milone, Igor Isajlović, Nedo Vrgoč, Michela Martinelli.

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
