## [Decision Letter · Decision Letter 0]

7 Apr 2022

PONE-D-22-00063Accounting for environmental and fishery management factors when standardizing CPUE data from a scientific survey: a case study for Nephrops norvegicus in the Pomo Pits area (Central Adriatic Sea)PLOS ONE

Dear Dr. Chiarini,

Thank you for submitting your manuscript to PLOS ONE. After careful consideration, we feel that it has merit but does not fully meet PLOS ONE’s publication criteria as it currently stands. Therefore, we invite you to submit a revised version of the manuscript that addresses the points raised during the review process.

All reviewers point out that the manuscript show some merits, However, I agree with the reviewers opinion, that the work is poorly presented/explained. In particular the reviewers highlight some aspects unclear about the modelling approach used. The results also require a thorough review as well as in the discussion section (see the specific comments of reviewers). Please, carefully address all reviewers comments in the revised version of the manuscript. Taking into account that this is a resubmission of the manuscript I expect to receive a much better version of the article otherwise I will be obliged to not accept it for publication

We look forward to receiving your revised manuscript.

Kind regards,

Pierluigi Carbonara, PhD

Academic Editor

PLOS ONE

https://journals.plos.org/plosone/s/file?id=ba62/PLOSOne_formatting_sample_title_authors_affiliations.pdf".

4. We note that [Figures 1, 4-7] in your submission contain [map/satellite] images which may be copyrighted. All PLOS content is published under the Creative Commons Attribution License (CC BY 4.0), which means that the manuscript, images, and Supporting Information files will be freely available online, and any third party is permitted to access, download, copy, distribute, and use these materials in any way, even commercially, with proper attribution. For these reasons, we cannot publish previously copyrighted maps or satellite images created using proprietary data, such as Google software (Google Maps, Street View, and Earth). For more information, see our copyright guidelines: http://journals.plos.org/plosone/s/licenses-and-copyright.

a. You may seek permission from the original copyright holder of [Figures 1, 4-7] to publish the content specifically under the CC BY 4.0 license. 

Natural Earth (public domain): http://www.naturalearthdata.com/.

Reviewers' comments:

Reviewer's Responses to Questions

**Comments to the Author**

1. Is the manuscript technically sound, and do the data support the conclusions?

Reviewer #1: No

Reviewer #2: Yes

Reviewer #3: Partly

2. Has the statistical analysis been performed appropriately and rigorously? 

Reviewer #1: No

Reviewer #2: Yes

Reviewer #3: N/A

3. Have the authors made all data underlying the findings in their manuscript fully available?

Reviewer #1: No

Reviewer #2: No

Reviewer #3: Yes

4. Is the manuscript presented in an intelligible fashion and written in standard English?

Reviewer #1: Yes

Reviewer #2: Yes

Reviewer #3: Yes

5. Review Comments to the Author

Reviewer #1: This manuscript presents GAM-based models of Nephrops survey data, identifying variables associated with areal pattern in density and/or biomass. Some details of the methods are confusing or not explained. Importantly, the concept of standardization is not well thought through and could be better explained. This might also be achieved with a clearer example(s) of how unstandardized data are used currently and what quantitative difference is made by using a multiple variable model.

The positioning of the analysis as ‘standardization’ is possibly appropriate, but not carefully presented. Standardization might usually apply to a correction for some sort of sampling variable (e.g., different tow lengths standardized to a constant length). The authors present an analysis of factors effecting the response variables, but this does not seem to be a standardization. It would help to be more explicit and compare the fitted surface to a not-standardized equivalent. For example multiplying the average density or biomass by the total area of the fishery (= non-standardized for any covariates).

While it may help to standardize variation in space (by providing more data), the use of ‘year’ is problematic as part of a standardization. The point of CPUE is often to follow changes over time (in the fishery of interest). The changes with year identified in the analysis presented in the manuscript are interesting, but the phrase ‘standardizing by year’ doesn’t make sense in the conventional use of CPUE data. This is also reflected by other comments highlighting the confusion between abundance and catchability. The revision does not do a good job of considering the wisdom of mixing year (abundance trend) and time of day (catchability change) in the analysis.

The methods are unclear is several respects:

1) The datasets were merged (line 233). How were they merged? If they were merged, why are there two response variables (density and biomass).

2) Analyses of the UWTV data have a factor for time of day. The text implies that trawl protocols were the same (line 214), so that videos were taken at sunset and sunrise. However, videos count burrows. How can the density of structures like burrows logically vary between sunrise and sunset (Table 2)? This does not make sense.

3) The partial effects plots are very similar (figure 2 and 3). If the data were merged as stated, why are there separate plots? If the data were not merged, why are the plots so similar?

4) The same point apples to figures 4 – 7. It is not clear what data were used to fit the models. They look very similar plots. Is this (for example) extrapolation of a spring model with autumn data or snapshots of a general model produced from all data? Why do we need 4 plots – this point should be made clear or fewer plots presented (but again, the sources of data need to be clear).

5) The conversion from burrows to numbers per km2 does not appear to be explained. Burrows are not subject to debates about catchability, although the (unstated) conversion to density will have some assumptions.

6) The models are fit with “gamma distribution with logarithmic link function” (line 251), but this is contradicted by “logit link function” on line 301.

7) Symmetric standard errors are not appropriate with skewed data (e.g., figure 8)

8) R2 is (line 433) is not a good measure for model fit with skewed data. R2 is also not a good measure of model performance in this sort of exploratory multiple variable setting. The average r2 (if it can be calculated given the skewed data) from cross validations is a better measure of model fit.

9) Why are observations missing from the partial plots for year in Figures 2 and 3?

Reviewer #2: Chiarini et al. analysed data from two seasonal surveys for addressing the interesting aim to investigate how some environmental variables and different management strategies can influence abundance and distribution of Nephrops norvegicus and to produce standardized CPUEs. The dataset comes from two surveys performed in an area, the Pomo/Jabuka Pits, subjected to different management measures in time and space. The authors used GAM models and found that bathymetry, fishing pressure, dissolved oxygen and salinity proved to be significant drivers influencing catch distribution for N. norvegicus in the area.

The new manuscript is largely well written and easy to follow, and I appreciated the authors' effort to address all the raised questions and concerns from the previous referees.

Used methodology appears clearly explained and their choice appropriate for the analysis done.

The final discussion chapter appears exhaustive, with a good selection of reference literature.

I only have some very few minor questions/comments listed below.

Figure/table captions: please check all captions and relative citation in the text according to the formatting guidelines. (i.e. Cite figures as “Fig 1”, “Fig 2”, use bold type for the figure titles, etc.)

Line 213. It is not clear: does “However, the planned hauls were equally allocated among areas…” refer to the performed hauls in the years when not all hauls could be done? If it is so, please change it with “However, the remaining performed hauls were equally allocated among areas…”. Instead, if the sentence refers to all the planned hauls in general, indicating that the management level is somehow taken into consideration in the planning of the two surveys, delete “however”.

Line 291. “for both for density and biomass CPUE” change in “for both density and biomass CPUE”

Line 398 “The Tukey post hoc test evidenced that “modINITIAL” was more statistically sound (lower AIC values) than “modNOE” and “modNOM””. It is not well clear why the Tukey test showed that the modINITIAL resulted more statistically sound: the test performs a pairwise comparison and showed a significant difference among all the pairs of models. The choose of the best model seems to be done by the lowest mean AIC values from the cross validation (Table S5).

Line 431. The tables should be tables 2 and 3 (not 1 and 2)

Line 438. The tables should be tables 2 and 3

Line 440 Table 2

Line 447 Table 3

Line 457-462. The sentence is too long, please, divide it.

Line 462. These should be table 2 and 3

Line 489 “after which level it...” change in “after which it…”

Line 541 Correct the figure caption with “…The "UWTV" time series is on the left (a and c panels), the "ScamPo" time series is on the right” (b and d panels)

Line 550 “…increment of about 67%”: please verify the percentage, from 24.9 to 37.33 it seems more an increase of about 50%

Reviewer #3: The study by Chiarini et al., presents an analysis to standardize N. norvegicus CPUE from 2 surveys from 2012 to 2019 using GAM models. In particular the authors i) investigate how the environmental variables and the different management strategies can influence abundance and distribution of Nephrops, ii) produce standardized CPUEs, to be used as reliable inputs for population dynamic and ecosystem models. The data are coming from two seasonal surveys, the areas associated with each survey being under different management measures.

I found the objectives of the study very interesting, but I have major comments about how the method and the results can addressed the main objectives of this paper. I then recommend major revision and would be very interested in seeing their responses to my questions/comments.

Major comments

Major Comments 1 :It was very difficult to me to figure out the spatiotemporal dimensions of the data and the model :

Major Comments 1.a :Data: it was difficult to me to understand the resolution of the observed data :

- Is there a sampling grid ? are the bubble points in Figure S1-S4 at the same locations than the green triangles in Figure 1 ?

- When I look at figure S1-S4 : are there the raw observed data ? Depending on years some hauls seem to have zero densities (example in FigS1 there in 2017 in the northwest there are positive catches but then at the same point there is not catches). So, this makes me think that some observations (CPUE) are equal to zero. I am then surprised that the models do not account for catches=0 (by using a zero inflated approaches or using a negative binomial distribution). Especially, there is no sentences or explanations about how the model deal with zero (the percentage of zero for each year and each survey ?)

Major Comments 1.b :Model: I find very interesting to account for different levels of time within the model: year/week/date of day. However, it is difficult to me to understand the dimensions (i.e time and space) of the model. If the model accounts for both weeks and years, it is complicated to me to understand the ecological meaning of the effects of bottom temperature (same for oxy, Sal). According to my understanding, the model (eq1) estimates here an average effect of temperature (and oxy, Sal) on CPUE. But because the authors are not providing any hypothesis about the mechanisms tested, I am a bit lost. The authors have data (and the model) to estimate the seasonal changes but here they use those data to estimate an average effect (which can be interpretate I guess as a niche effect/habitat suitability). So at the end of the day, I am curious to know why putting so much efforts in collecting so many great data (I am wondering if yearly averaged temperature (oxygen/Sal) outputs from oceanographic models would have provided the same kind of information for example?).

Again, I think that if the underlying mechanisms linking the covariates (in time (year, week, date of day) and space) and CPUE would have been better defined, I think it would have helped me a lot to better understand what is going on with the model.

Major Comments 1.c :Covariates: Which covariates are varying in space and time ? I found figure S5 and S10 very useful to understand the spatiotemporal variation in environmental covariates. However, it is difficult to me to picture exactly how vary the fishery covariates. My understanding is that they are spatially defined (Zone A, B, C) and temporally defined? Can you please provide a table or a plot of those fishery covariates? It will help the reader to better understand how those covariates vary in time and space.

Major Comments 2 : Because one of the objective of the paper is “to produce standardized CPUE to be used […] for population dynamic (line 148-150), I would have expected more information about the current population dynamic used for this species : actually is there one ? if yes why does it need new standardized CPUE ? how different are the new standardized CPUE produced in this study with the current CPUE used?

Because now the standardized CPUE are defined for each season, I am wondering if they will be able to be implemented in the population dynamic model, is this model seasonally defined ?

Regarding the objectives of the paper “to produce standardized CPUE to be used […] for population dynamic (line 148-150)” , I think we are missing some context/background about how standardized CPUE are/could be used in the population dynamics of this species

Major Comments 3 :To account for spatial variability, I am curious to understand why the authors planned to account independently for latitude and longitude and not for the interaction which I guess could be a better proxy of the spatial location (Potts and Rose, 2018).

Major Comments 4 : I do not understand why all the analysis are repeated for both abundance density (N/km2) and biomass density (kg/km2) (the authors call it biomass it is biomass/area). All tables and figures are repeated without a clear message.

Major Comments 5: In the results section, it was difficult to me, to extract a clear message and arguments leading to meet the objectives of this study. For example, line 495-497 (also see line 531-534), this sentence presents results and explains how those results have been compiled (which according to me should be in the M&M). As a consequence, I think this makes the ‘results section’ a bit unclear. Also, it is really not clear to me (see comments 4) why there is so many comparisons and figures between densities and biomass. I do not understand how this help to meet the objectives of this paper.

Major Comments 6 : I think the discussion is missing a strong paragraph about the method used and other potential methods. Especially because the authors highlighted that GAMs are the best for this kind of study “in this study GAMs were selected as the best”. Which I think is not false but at least should be discussed ( it is more complicated than just saying GAMs are the best”)

In the introduction the authors highlight that considering for spatial autocorrelation is important (line 103). But in the method proposed in this study, they do not account for it (or maybe just through the smoother term for Latitude and I am OK with that). But I think it would be interesting to discuss this point in the discussion section. What information are we missing when not considering spatial autocorrelation, how can we account for this spatial autocorrelation better (Thorson, 2019; Brodie et al., 2020)

Also an important element should be discussed : what are the best models to answer those two questions : (I) investigate how the environmental variables and the different management strategies can influence abundance and distribution of Nephrops, (II) produce standardized CPUEs. One model can highly perform for question (I) and poorly perform for question II and inversely. Please see the very interesting discussion in (Brodie et al., 2020) “if model purpose is to best explain the environmental processes underlying species distributions, then BRTs and GAMs would be a best choice” , “model purpose may be to obtain robust abundance indices with associated standard errors, in which case VAST would be the best choice. In such cases, the most accurate abundance indices could be included in stock assessment”.

Intermediate/Minor comments

line 67: I do not know what “strategy changes” is in such a global context

line 116 : what does “etc…” mean here ? I think it is important to define the main ecological process driving the population dynamic of Nephrops, so here “etc…” seems irrelevant to me.

Line 117-119 : Can we know more about the ecological mechanisms linking oxygen, salinity and bottom temperature to nephrod life cycle (or are they random correlations ?)

Line 127: by “differences over time”, do the authors mean, time of the day, and season and year ? It could help to better understand the temporal resolution of the study if the authors could be more precise

Line 222 -229 : Can the authors explicitly define the environmental data ? here I can only guess : temperature (bottom or surface ?) and oxygen (bottom or surface ?). What about salinity which is then use in the model.

Line 231 : what are the biotic and abiotic variables.

Line 250 : why a gamma and not a lognormal for example. I am not sure to understand the justification for a gamma distribution here.

Line 253 : I would say that the log-link function is mostly useful because the effects on log(CPUE) are then additive and it is therefore easier to understand/interpret

Line 254-258 : Here again, I think it would give more power to the study if we could understand the underlying mechanisms linking CPUE and all covariates (Y, X, D, Yr, week, ToD, BT, Sal, Oxy, Fishery)

Line 260 : I do not know a lot about this fishery : are the restriction on fishery only occurring at the Year level or also at a seasonal level (for example the buffer zone). I am trying to understand here if the interaction between fishery and time should be only fishery*year or also fishery*weeks.

Line 276: Why “few”? there are 3 covariates: oxygen, salinity and temperature. They represent an important amount of information, so I will not use the word “few”.

Line 286 : with which other covariates, longitude is correlated : I think it might be an interesting information to understand which covariates carry the same amount of information.

Line 291 : At this point of the manuscript I do not understand why the authors are interested in both density and biomass. What are they investigating differently, when using biomass or density ?

Line 294 : It is difficult to me to understand the dimensions (spatial and temporal) of the variable CPUE. Here CPUE are defined for each latitude, longitude, week, years, others ? please define better CPUE, maybe add subscript for space and time ?

Line 300: is it a yearly, and/or spatial and/or weekly error term? Again, I would recommend to better defined the dimension of the variables in the model

Line 313 – 320 : How do we know that the terms Y, D, week and Yr that are common to equation (2) and equation (3), can explain variations in CPUE (are they significant ?).

So here a suggestion : I am wondering if first, it would not make more sense to figure out a model M0 with no environmental and fishery covariates, with only significant effects of covariates (Y, D week, ToD) on CPUE.

And then, once model M0 is validated, define model M2 (M0 + environmental covariates) et M3 (M0+fishery covariates) to figure out if environmental covariates (M2) explain more OR less variability than fishery covariates (M3). And finally, M4 will be M0 + Env.cov +fishery.cov (with only the covariates having a significant effects), which will be used to produce standardized CPUE.

Also, if the authors want to answer the question they asked in Introduction: “investigate how the environmental variables and the different management strategies can influence abundance and distribution of Nephrops” I think they should be able to estimate the part of variance explained by each covariate, which will help them to conclude (i) about how the selected environmental influence abundance and (ii) discuss the underlying mechanisms.

Line 330: I am not sur to know what is “the training data set”

Line 346-349 : I am not sure to understand this sentence : why using all data at hand if for example the model selection suggests not using the covariate week. I am maybe missing something here, but this sentence is confusing to me.

Line 355 : what are “means of the final models” : means of which variable ? of the estimated effect of covariate ? across space ?

Line 384-388 : One of the objective of the paper is to “produce standardized CPUEs, to be used as reliable inputs for population dynamic and ecosystem models” (line 149). But does the population dynamic model has a seasonal resolution? if not, I do not think that the authors can say they provided information to feed the population dynamic. Because here the authors provided standardized CPUE for each survey. I think that most of the readers would have expected an annual global index of abundance from this study, and maybe a plot comparing the previous and the new standardized CPUE.

Line 600:601: “ in this study GAMs were selected as the best ….” it suggested that the authors have tested other models, did they ?

Also yes GAM are great but other models used for species distribution models can been used to fill the gaps of a time-serie by using autoregressive processes (Thorson, 2019; Anderson et al., 2022).

References

Anderson, S. C., Ward, E. J., English, P. A., and Barnett, L. A. K. 2022. sdmTMB: an R package for fast, flexible, and user-friendly generalized linear mixed effects models with spatial and spatiotemporal random fields. preprint. Ecology. http://biorxiv.org/lookup/doi/10.1101/2022.03.24.485545 (Accessed 29 March 2022).

Brodie, S. J., Thorson, J. T., Carroll, G., Hazen, E. L., Bograd, S., Haltuch, M. A., Holsman, K. K., et al. 2020. Trade-offs in covariate selection for species distribution models: a methodological comparison. Ecography, 43: 11–24.

Potts, S. E., and Rose, K. A. 2018. Evaluation of GLM and GAM for estimating population indices from fishery independent surveys. Fisheries Research, 208: 167–178.

Thorson, J. T. 2019. Guidance for decisions using the Vector Autoregressive Spatio-Temporal (VAST) package in stock, ecosystem, habitat and climate assessments. Fisheries Research, 210: 143–161.

6. PLOS authors have the option to publish the peer review history of their article (what does this mean?). If published, this will include your full peer review and any attached files.

Reviewer #1: No

Reviewer #2: No

Reviewer #3: No

---

## [Author Response · Author response to Decision Letter 0]

8 May 2022

please find within “Chiarini et al. (PlosONE)_Response to Reviewers.docx” in red characters point-by-point replies to all Reviewers’ comments and suggestions.

---

## [Decision Letter · Decision Letter 1]

10 Jun 2022

PONE-D-22-00063R1Accounting for environmental and fishery management factors when standardizing CPUE data from a scientific survey: A case study for Nephrops norvegicus in the Pomo Pits area (Central Adriatic Sea)PLOS ONE

Dear Dr. Chiarini,

Thank you for submitting your manuscript to PLOS ONE. After careful consideration, we feel that it has merit but does not fully meet PLOS ONE’s publication criteria as it currently stands. Therefore, we invite you to submit a revised version of the manuscript that addresses the points raised during the review process.

We look forward to receiving your revised manuscript.

Kind regards,

Pierluigi Carbonara, PhD

Academic Editor

PLOS ONE

Journal Requirements:

Reviewers' comments:

Reviewer's Responses to Questions

**Comments to the Author**

1. If the authors have adequately addressed your comments raised in a previous round of review and you feel that this manuscript is now acceptable for publication, you may indicate that here to bypass the “Comments to the Author” section, enter your conflict of interest statement in the “Confidential to Editor” section, and submit your "Accept" recommendation.

Reviewer #2: All comments have been addressed

Reviewer #3: All comments have been addressed

2. Is the manuscript technically sound, and do the data support the conclusions?

Reviewer #2: Yes

Reviewer #3: Yes

3. Has the statistical analysis been performed appropriately and rigorously? 

Reviewer #2: Yes

Reviewer #3: Yes

4. Have the authors made all data underlying the findings in their manuscript fully available?

Reviewer #2: Yes

Reviewer #3: No

5. Is the manuscript presented in an intelligible fashion and written in standard English?

Reviewer #2: Yes

Reviewer #3: Yes

6. Review Comments to the Author

Reviewer #2: In this study the authors applied GAMs to produce specific standardized indices, in number and weight, accounting for the effects of both environmental variables and fishery management actions. I believe that this study can be very interesting due to the need to standardize survey data, an increasing necessity mainly in the pandemic period where a lot of surveys have often been made but in a wrong period.

The authors have done a very comprehensive job of revising their manuscript and have addressed all previous concerns from me and the other referees. Although I enjoyed the previous version as well, the current version shows some more improvements making it better matching the required journal standards.

For this reason, I recommend acceptance for this manuscript as it is.

Reviewer #3: The study by Chiarini et al., presents an analysis to standardize N. norvegicus CPUE from 2 surveys from 2012 to 2019 using GAM models. In particular the authors i) investigate how the environmental variables and the different management strategies can influence abundance and distribution of Nephrops, ii) produce standardized CPUEs, to be used as reliable inputs for population dynamic and ecosystem models. The data are coming from two seasonal surveys, the areas associated with each survey being under different management measures.

I found the objectives of the study interesting. The questions addressed in this manuscript seem to be very important for the ecosystem studied (Pomo Pits area (Central Adriatic Sea)).

I appreciated that the authors answered and accounted for all my comments, that is why I recommend minor revision. So with the exception of some comments/questions described below, I found the manuscript’s study design robust and its conclusions fair. Below I sum up my main remaining comments that I would like the authors to answers.

Line 268-269 : 2.6 hauls removed per year : Which % does it represent per year because I guess it is represent a different % between 2012 (17 hauls) and 2013 (10 hauls)

Line 430-437 : I am wondering : If the objective is to provide indices to feed stock assessment model, instead of averaging across the predicting grid, is it not more appropriate to multiplicate each density by the area of the grid and then sum it across the grid to have a the total abundance (biomass) of the area

Line 444-447 : I am not familiar with an ANOVA test of the AIC values. Is there any paper justifying this approach? To my knowledge if the difference between AIC is 5, the model with the lowest AIC is considered better.

Line 643-649 : I will move this information, in the paragraph “purpose of the study”. Because I think the reader does not understand why he goes through all those repeated analysis for both abundance and biomass. And results seem very similar for abundance and biomass ( I feel that it is only on line 618, we really identify there is a difference). In the discussion paragraph, I will rather say if there is discrepancy or not (and sum up what are the main differences), instead of saying why the authors carried out standardization on both abundance and biomass (the “why” should be in Introduction or Method)

Line 828 : It is more a comment than a request. I am curious to have the authors point of view. Because depending on the years, the sampling (number of hauls) is not homogeneous (in year 2012, there are 17 hauls whereas in 2013 there is a 10 hauls) can we think that the year effect in the model is confounded with a sampling effect ?

7. PLOS authors have the option to publish the peer review history of their article (what does this mean?). If published, this will include your full peer review and any attached files.

Reviewer #2: No

Reviewer #3: No

---

## [Author Response · Author response to Decision Letter 1]

13 Jun 2022

please find replies to reviewers and editors comments within “Cover Letter.docx” .

---

## [Editor Report · Decision Letter 2]

16 Jun 2022

Accounting for environmental and fishery management factors when standardizing CPUE data from a scientific survey: A case study for Nephrops norvegicus in the Pomo Pits area (Central Adriatic Sea)

PONE-D-22-00063R2

Dear Dr. Chiarini,

We’re pleased to inform you that your manuscript has been judged scientifically suitable for publication and will be formally accepted for publication once it meets all outstanding technical requirements.

Kind regards,

Pierluigi Carbonara, PhD

Academic Editor

PLOS ONE

---

## [Editor Report · Acceptance letter]

22 Jun 2022

PONE-D-22-00063R2 

Accounting for environmental and fishery management factors when standardizing CPUE data from a scientific survey: A case study for *Nephrops norvegicus* in the Pomo Pits area (Central Adriatic Sea) 

Dear Dr. CHIARINI:

I'm pleased to inform you that your manuscript has been deemed suitable for publication in PLOS ONE. Congratulations! Your manuscript is now with our production department. 

Kind regards, 

on behalf of

Dr. Pierluigi Carbonara 

Academic Editor

PLOS ONE